# First evidence of a monodominant (*Englerodendron*, Amherstieae, Detarioideae, Leguminosae) tropical moist forest from the early Miocene (21.73 Ma) of Ethiopia

**Aaron D. Pan**[1,2ʘ]*, **Bonnie F. Jacobs**[3ʘ], **Rosemary T. Bush**[4‡], **Manuel de la Estrella**[5‡], **Friðgeir Grímsson**[6ʘ], **Patrick S. Herendeen**[7ʘ], **Xander M. van der Burgt**[8‡], **Ellen D. Currano**[9ʘ]

**1** Museum of Texas Tech University, Texas Tech University, Lubbock, Texas, United States of America, **2** Botanical Research Institute of Texas, Fort Worth, Texas, United States of America, **3** Roy M. Huffington Department of Earth Sciences, Southern Methodist University, Dallas, Texas, United States of America, **4** Department of Earth and Planetary Sciences, Northwestern University, Evanston, Illinois, United States of America, **5** Departamento de Botánica, Ecología y Fisiología Vegetal, Celestino Mutis, Campus de Rabanales, Universidad de Córdoba, Córdoba, Spain, **6** Department of Botany and Biodiversity Research, University of Vienna, Vienna, Austria, **7** Division of Plant Science and Conservation, Chicago Botanic Garden, Glencoe, Illinois, United States of America, **8** Africa Team, Herbarium, Royal Botanic Gardens, Kew, Richmond, United Kingdom, **9** Department of Botany and Geology and Geophysics, University of Wyoming, Laramie, Wyoming, United States of America

ʘ These authors contributed equally to this work.
‡ RTB, ME and XMB also contributed equally to this work.
* aaron.pan@ttu.edu

## Abstract

Many tropical wet forests are species-rich and have relatively even species frequency distributions. But, dominance by a single canopy species can also occur in tropical wet climates and can remain stable for centuries. These are uncommon globally, with the African wet tropics supporting more such communities than the Neotropics or Southeast Asia. Differences in regional evolutionary histories are implied by biogeography: most of Africa's monodominance-forming species are Amherstieae-tribe legumes; monodominance in Neotropical forests occur among diverse taxonomic groups, often legumes, but rarely Amherstieae, and monodominance in Southeast Asian forests occurs mostly among Dipterocarpaceae species. African monodominant forests have been characterized ecologically and taxonomically, but their deep-time history is unknown despite their significant presence and bottom-up ecological influence on diversity. Herein we describe fossil leaflets of *Englerodendron mulugetanum* sp. nov., an extinct species of the extant genus *Englerodendron* (Berlinia Clade, Amherstieae, Detarioideae) from the 21.73 Ma Mush Valley site in Ethiopia. We also document a detailed study of associated legume pollen, which originate from a single taxon sharing characters with more than one extant descendant. Taxonomically, the pollen is most comparable to that from some extant *Englerodendron* species and supports a likely affiliation with the *Englerodendron* macrofossils. The Mush Valley site provides the first fossil evidence of a monodominant tropical forest in Africa as represented by leaflets and pollen. Previous studies documented >2400 leaves and leaflets from localities at six

**Data Availability Statement:** All relevant data are within the paper and its Supporting Information files.

**Funding:** Funded by U.S. National Science Foundation (https://www.nsf.gov/) grants EAR 1053549 (BFJ) and EAR 1052478 (EDC), National Geographic Society Grant (https://www.nationalgeographic.org/society/grants-and-investments/) CRE 8816-10 (EDC), and the Austrian Science Fund (https://www.fwf.ac.at/en/), FWF, P29501-B25 (FG). The funders had no role in study design, data collection and analysis, decision to publish, or preparation of the manuscript.

**Competing interests:** The authors have declared that no competing interests exist.

stratigraphic levels spanning 50,000–60,000 years of nearly continuous deposition within seven meters of section; all but the basal level contain ≥ 50% *E. mulugetanum* leaflets. Modern leaf litter studies in African mixed vs. monodominant forests indicates the likelihood of monodominance in the forests that surrounded the Mush paleolake, particularly after the basal level. Thus, we provide an early case for monodominance within the Amherstieae legumes in Africa.

## Introduction

Modern moist and wet primary tropical forest formations are typically characterized by high to extremely high levels of biodiversity and can boast as many as 300 tree species within a single hectare [1, 2]. More unusual are natural moist and wet tropical forest formations that are dominated by a single tree species, often making up from 50% to nearly 100% of the canopy trees [3, 4]. While this phenomenon can be found in the tropics of Africa, the New World, and Asia, there are clear biogeographic differences among the three tropical regions, in part caused by the occurrence of monodominant species in at least 24 families and 51 genera. The majority of monodominant forests in the Neotropics and Africa are composed of species in the legume family (Leguminosae), but compared with Africa and Asia, Neotropical monodominant-forming forest taxa are more diverse; this is true whether considering all families or the diversity of subfamilies and tribes among legume dominants (Table 1) [3, 5–7]. African monodominant tropical forests are nearly always characterized by legume species in the tribe Amherstieae of the subfamily Detarioideae, many of which occur in the monophyletic and African endemic Berlinia Clade (Table 1) [8, 9]. A number of genera and species (even those not forming monodominant forest formations) within the Berlinia Clade have a tendency to form monotypic stands or are noted as often becoming gregarious where found [10]–thus, this African clade may have a propensity for forest ecological dominance. Tropical Asia is well-known and unique for the importance and Neogene diversity of the Dipterocarpaceae, and its monodominant forests are composed of species primarily in this family (Table 1) [11, 12].

In addition to its biogeographic distinctiveness, Africa is unique among tropical regions with regard to the large areal extent of moist and wet monodominant forests [7, 13], which collectively occupy thousands of km$^2$ [5, 14–16]. They can be found in the Congo Basin to Cameroon and Nigeria and are dominated by, for example, *Gilbertiodendron dewevrei* (De Wild.) J. Léonard, 1952, *Julbernardia seretii* (De Wild.) Troupin, 1950, and *Brachystegia laurentii* (De Wild.) Louis ex J.Léonard, 1952 [5, 14–16]. A number of causal mechanisms and species traits have been called upon to explain why in some circumstances monodominance emerges and then can persist for centuries. These include deep leaf litter and slow litter decomposition, large seed size, nutrient-poor soils, the presence of ectomycorrhizal symbionts, competitive advantages among the dominant taxa, and both shade and sun tolerance [4, 17–19]. Detarioideae legumes have a long evolutionary history in Africa and the Neotropics dating to the early Cenozoic [20, 21], one might expect monodominance to have occurred in the deep past, perhaps even in South America where this phenomenon is relatively rare today. None has been identified to date, leaving undocumented the origins and biogeographic history of monodominant tropical wet forests.

In this paper, we describe an early Miocene taxon from Ethiopia (formerly referred to as "Legume 1" [22, 23]) and assign it to the Berlinia Clade. Previous work on the paleoecology of the locality that produced the new taxon found strong support for a forest community

**Table 1. Monodominant tropical forests of the world and their dominant species component.**

| MONODOMINANT TROPICAL FOREST | SPECIES | HIGHER TAXONOMIC RANKINGS | CONTINENT | CITATION |
|---|---|---|---|---|
| | *Aucoumea klaineana* | Burseraceae | AFRICA | [66] |
| | *Terminalia superba* | Combretaceae | AFRICA | [66] |
| | *Aphanocalyx microphyllus* | **Leguminosae: Detarioideae: Amherstieae** | AFRICA | [61] |
| | *Brachystegia laurentii* | **Leguminosae: Detarioideae: Amherstieae** | AFRICA | [5] |
| | *Cynometra alexandri* | **Leguminosae: Detarioideae: Amherstieae** | AFRICA | [5] |
| Mbau (*Gilbertiodendron dewevrei*) forest | *Gilbertiodendron dewevrei* | **Leguminosae: Detarioideae: Amherstieae** | AFRICA | [5, 6] |
| *Julbernardia seretii* | *Julbernardia seretii* | **Leguminosae: Detarioideae: Amherstieae** | AFRICA | [5] |
| | *Microberlinia bisulcata* | **Leguminosae: Detarioideae: Amherstieae** | AFRICA | [62] |
| | *Talbotiella gentii* | **Leguminosae: Detarioideae: Amherstieae** | AFRICA | [92] |
| | *Tetraberlinia bifoliolata* | **Leguminosae: Detarioideae: Amherstieae** | AFRICA | [62] |
| | *Tetraberlinia korupensis* | **Leguminosae: Detarioideae: Amherstieae** | AFRICA | [62] |
| | *Tetraberlinia tubmaniana* | **Leguminosae: Detarioideae: Amherstieae** | AFRICA | [93] |
| | *Gilletiodendron glandulosum* | Leguminosae: Detarioideae: Detarieae | AFRICA | [68] |
| | *Musanga cecropioides* | Urticaceae | AFRICA | [94] |
| Kapur forest | *Dryobalanops aromatica* | Dipterocarpaceae | ASIA (Malesia) | [95] |
| | *Parashorea malaanonan* | Dipterocarpaceae | ASIA (Malesia) | [96] |
| | *Shorea albida* | Dipterocarpaceae | ASIA (Malesia) | [97] |
| | *Shorea curtisii* | Dipterocarpaceae | ASIA (Malesia) | [98] |
| Ulin forest | *Eusideroxylon zwageri* | Lauraceae | ASIA (Malesia) | [99] |
| | *Palaquium gutta* | Sapotaceae | ASIA (Malesia) | [100] |
| | *Dimorphocalyx glabellus* | Euphorbiaceae | ASIA (South Asia: India) | [101] |
| | *Strychnos nux-vomica* | Loganiaceae | ASIA (South Asia: India) | [101] |
| | *Tricalysia sphaerocarpa* | Rubiaceae | ASIA (South Asia: India) | [101] |
| Sal forest | *Shorea robusta* | Dipterocarpaceae | ASIA (South Asia: India-Bangladesh) | [102] |
| | *Hopea ferrera* | Dipterocarpaceae | ASIA (Southeast Asia: NE Thailand) | [103] |
| | *Backhousia bancroftii* | Myrtaceae | AUSTRALIA | [104] |
| | *Celaenodendron mexicanum* | Euphorbiaceae | MESOAMERICA | [105] |
| | *Quercus oleoides* | Fagaceae | MESOAMERICA | [106] |
| | *Pisonia grandis* | Nyctaginaceae | OCEANIA (Micronesia) | [107] |
| Ōhiʻa forest | *Metrosideros polymorpha* | Myrtaceae | OCEANIA (Polynesia: Hawaii) | [108] |
| | *Oxandra polyandra* | Annonaceae | SOUTH AMERICA | [2] |
| | *Astrocaryum macrocalyx* | Arecaceae: Arecoideae: Cocoseae | SOUTH AMERICA | [2] |
| | *Astrocaryum murumuru* | Arecaceae: Arecoideae: Cocoseae | SOUTH AMERICA | [2] |
| | *Attalea speciosa* | Arecaceae: Arecoideae: Cocoseae | SOUTH AMERICA | [2] |
| | *Euterpe oleracea* | Arecaceae: Arecoideae: Euterpeae | SOUTH AMERICA | [109] |
| | *Mauritia flexuosa* | Arecaceae: Calamoideae: Lepidocaryeae | SOUTH AMERICA | [2] |
| | *Jacaranda densicoma* | Bignoniaceae | SOUTH AMERICA | [2, 110] |
| | *Tabebuia aurea* | Bignoniaceae | SOUTH AMERICA | [2] |
| | *Micrandra glabra* | Euphorbiaceae | SOUTH AMERICA | [2] |
| | *Mora gonggrijpii* | Leguminosae: Caesalpinoideae | SOUTH AMERICA | [2] |
| | *Mora oleifera* | Leguminosae: Caesalpinoideae | SOUTH AMERICA | [2] |
| | *Tachigali vaupesiana* | Leguminosae: Caesalpinoideae | SOUTH AMERICA | [2] |

(*Continued*)

**Table 1.** (Continued)

| MONODOMINANT TROPICAL FOREST | SPECIES | HIGHER TAXONOMIC RANKINGS | CONTINENT | CITATION |
|---|---|---|---|---|
| | *Mora excelsa* | Leguminosae: Caesalpinoideae: *Dimorphandra* Group A | SOUTH AMERICA | [2] |
| | *Dicymbe corymbosa* | **Leguminosae: Detarioideae: Amherstieae** | SOUTH AMERICA | [111] |
| | *Eperua falcata* | Leguminosae: Detarioideae: Detarieae | SOUTH AMERICA | [2] |
| Jauacaná caatinga forest | *Eperua leucantha* | Leguminosae: Detarioideae: Detarieae | SOUTH AMERICA | [2] |
| Caatinga Forest (Jebarú) | *Eperua purpurea* | Leguminosae: Detarioideae: Detarieae | SOUTH AMERICA | [2] |
| | *Peltogyne gracilipes* | Leguminosae: Detarioideae: Detarieae | SOUTH AMERICA | [112] |
| cativales' | *Prioria copaifera* | Leguminosae: Detarioideae: Detarieae | SOUTH AMERICA | [113] |
| | *Machaerium hirtum* | Leguminosae: Faboideae: Dalbergieae | SOUTH AMERICA | [2] |
| | *Spirotropis longifolia* | Leguminosae: Faboideae: Ormosieae | SOUTH AMERICA | [114] |
| | *Pentacletha macroloba* | Leguminosae: Mimosoideae: | SOUTH AMERICA | [115] |
| | *Vitex cymosa* | Lamiaceae | SOUTH AMERICA | [2] |
| Igapó forest (black-water flooded forests) | *Eschweilera tenuifolia* | Lecythidaceae | SOUTH AMERICA | [116] |
| | *Lueheopsis hoehnei* | Malvaceae *sensu lato* | SOUTH AMERICA | [2] |
| | *Pachira nitida* | Malvaceae *sensu lato* | SOUTH AMERICA | [2] |
| | *Brosimum rubescens* | Moraceae | SOUTH AMERICA | [2] |
| | *Phyllanthus elsiae* | Phyllanthaceae | SOUTH AMERICA | [2] |
| | *Triplaris weigeltiana* | Polygonaceae | SOUTH AMERICA | [2] |
| | *Ruizterania retusa* | Vochysiaceae | SOUTH AMERICA | [2] |

dominated by legume taxa overall, noting the relative abundance of "Legume 1" in particular, which comprises 1289 specimens (53%) of all leaves collected and documented (2427). Its long-term abundance through seven vertical meters of sampling prompted us to take a closer look at the structural nature of the forest community represented by all the fossil leaf assemblages to determine if monodominance is or is not represented at this locality, with the keystone role played by a Berlinia Clade taxon as happens in Africa today. We also looked at the dispersed pollen flora from the same sediments as the macrofossils to explore if Berlinia clade pollen occurs in the Mush record and if similar "monodominant" signals are reflected by the palynoflora.

## Materials and methods

The Mush Valley fossil site is located in a region of the northwestern plateau of Ethiopia approximately 160 km northeast of Addis Ababa and between the towns of Debre Birhan and Debre Sina (9˚47'N, 39˚39' E) [22–25]. Abundant and diverse fossils including plants (leaf, wood, fruit, and seed compressions and pollen), vertebrates (rare mammals, anurans, and teleost fish compressions), and insect compressions are preserved in layers of lacustrine carbonaceous shale interbedded with thin layers of volcanic ash < ~3 cm thick [23–27]; one exceptional interbedded ash horizon (fossil-bearing Level C) is 10 cm thick. The fossiliferous shales are constrained in age by $^{206}Pb/^{238}U$ dates on zircons within two volcanic ashes, one that is stratigraphically immediately below them and one five additional meters below that. These yield the following ages for the upper and lower ashes, respectively: 21.733 ± 0.060 Ma and 21.736 ± 0.015 Ma [28].

Research, particularly field work and the collection and analysis of fossil plants and geologic specimens from the Upper and Lower Mush Valley, Amhara, Ethiopia, was permitted by the

Authority for Research and Conservation of Cultural Heritage (ARCCH), a governmental agency within the Ministry of Culture of Ethiopia. The necessary permits for multiple years of field work (2010–2013) resulting in the collection of the specimens referred to in this paper are on file at the national offices of the ARCCH in Addis Ababa, with copies to the regional office located in Debre Berhan. Preparation and description of fossils were permitted by the National Museum of Ethiopia, Addis Ababa, where the specimens are housed permanently. The project members complied with all relevant regulations. Additional information regarding the ethical, cultural, and scientific considerations specific to inclusivity in global research is included in S1 File.

Fossil leaves and leaflets were quarried from benches excavated such that a 5 cm thick shale unit could be removed in large blocks. To retard desiccation and specimen damage, the blocks were wrapped and transported in trunks to the National Museum of Ethiopia after each field season and left to dry during intervals of about six months at a time. Subsequent splitting, preparation, photography, and classification took place at the museum for all leaves or leaflets having $\geq$ 50% of the lamina preserved. Small, ~0.5 cm$^2$, cuticle samples were removed from leaves to aid in classification. Cuticle preparations and microscope slides from herbarium samples are housed in the Roy M. Huffington Department of Earth Sciences at Southern Methodist University, Dallas, TX, USA [23–25]. A total of 2427 leaves were classified (using morphotype names if not identified to taxon) from collections among two to three quarries at each of six stratigraphic levels (A-F) spanning 7 m of section [23]. A total of 49 morphotypes were classified, including "Legume 1", the most common taxon described and named in this paper (see also, [22]). Inventories of morphotype descriptions and number of specimens per quarry and level are provided in Currano et al. [23] (main text and supplemental materials). Frequency distributions, diversity measures, and isotopic measurements were analyzed to evaluate paleoecology and to identify any patterns of variation within and between stratigraphic levels, thus gaining a spatial and temporal view of plant community characteristics [22, 23, 27]. The estimated time represented by these shale deposits is approximately one to three centuries per 5 cm-thick quarry collection, and approximately 50–60 kyrs in total, on the basis of comparison with dated Holocene and Pleistocene lake cores [23].

Fossil specimens were compared to extant Leguminosae specimens from herbaria housed at the United States National Museum of Natural History (US), Missouri Botanical Garden (MO), Botanical Research Institute of Texas (BRIT), and, via high resolution digital images from the Muséum national d'Histoire naturelle (MNHN) and Global Plants on JSTOR (S2 File). Additional comparisons were based on published material (citations in RESULTS). Methods for preparation of herbarium and fossil leaflet cuticle and venation are provided in Herendeen & Dilcher [29] and Pan et al. [25], respectively. Herbarium leaf cuticle slides are housed at the Chicago Botanic Garden, IL, USA.

Sedimentary samples for pollen analysis were collected from each of the stratigraphic levels (A-F) alongside the macrofossils. Sedimentary rock samples were processed and fossil pollen grains extracted according to the method explained in Grímsson et al. [30]. The fossil pollen grains were investigated both by light microscopy (LM) and scanning electron microscopy (SEM) using the single grain method as described by Zetter [31] and Halbritter et al. [32; pp. 121–123]. Ten fossil pollen grains were measured and studied with LM and SEM. For practical reasons, the fossil pollen is classified as a morphotype (MT) named after the locality where the grains were found. The fossil pollen from Mush is described in detail based on LM and SEM observations and compared to pollen from related extant taxa. Part of the original sediment and remaining organic residues from palynological preparations and the SEM stubs bearing the fossil pollen are curated in the Division of Structural and Functional Botany at the University of Vienna, Austria.

Pollen from extant flower material (S3 File) from the Royal Botanic Gardens, Kew (K) was prepared according to the protocol outlined in Grímsson et al. [33, 34] and Halbritter et al. [32; pp. 103–105, Acetolysis the Fast Way]. Twenty-five pollen grains from each sample were measured and studied with LM and SEM. The pollen terminology follows Punt et al. [35; LM] and Halbritter et al. [32; SEM].

### Nomenclature

The electronic version of this article in Portable Document Format (PDF) in a work with an ISSN or ISBN will represent a published work according to the International Code of Nomenclature for algae, fungi, and plants, and hence the new names contained in the electronic publication of a PLOS ONE article are effectively published under the Code from the electronic edition alone, so there is no longer any need to provide printed copies.

The online version of this work is archived and available from the following digital repositories: PubMed Central and LOCKSS.

## Results

Leguminosae Jussieu 1789

Detarioideae Burmeist. 1837

Amherstieae Benth. 1840

*Englerodendron* Harms 1907

*Englerodendron mulugetanum* Pan, Jacobs, Bush, Estrella, Grímsson, Herendeen, Burgt et Currano, *sp. nov.*

**Holotype:** MU17-43 (Fig 1A). National Museum of Ethiopia, Addis Ababa

**Paratypes:** MU23-40(1C) #4 (Cuticle; Fig 3A), MU27-18#1 (Fig 1C), MU29-40B #1, MU41-15 (Cuticle; Fig 3B)

**Additional Material:** MU7-6#3, MU7-10, MU7-21#1, MU7-35-A5A#23, MU7-40#5, MU13-1, MU13-28#1, MU13-28#3, MU17-26A, MU18-16A#1, MU23-20, MU23-37#1 (Cuticle), MU23-59#1, and MU32-22#8, MU33-19, MU40-22 (S4 File).

**Diagnosis:** Paripinnate leaves, symmetrical leaflets, oppositely inserted with twisted and terete pulvinate short petiolules and acuminate leaflet tips. Secondary venation is eucamptodromous, often becoming brochidodromous towards the leaflet apex. Abaxial and adaxial cuticle with highly sinuous anticlinal cell walls, which also possess 'knob-like' thickenings. Stomata paracytic. Trichome bases present, but sparse.

**Description:** Modified and updated from Bush et al. [22]. Paripinnate leaves, possessing oppositely inserted symmetrical unlobed, untoothed (entire-margined), elliptic to slightly falcate leaflets with twisted and likely terete pulvinate short petiolules (Figs 1 and 2). Leaflet lamina generally range in size from circa 4.5 to 15.0 cm in length and 1.6 to 5.7 cm in width, typically with length: width ratios of 3:1 to 5:2. Leaflet midribs are canaliculate (Fig 2C). The leaflet apex is acute and possesses an acuminate 'drip' tip. The leaflet base is straight to rounded in shape and ranges from symmetrical to slightly asymmetrical [22; Fig 2A and 2C]. Primary venation is pinnate (Figs 1 and 2). Secondary venation is eucamptodromous, often becoming brochidodromous distally [22; Figs 1 and 2]. The secondary veins are irregularly spaced, have excurrent attachment, and generally possess uniform acute angles (Fig 2A and 2C). Leaflets typically possess (5 –) 6–7 (9–10) pairs of secondary veins. Inter-secondary veins are present, diverge from the midvein parallel to the secondary veins, are less than 50% of the

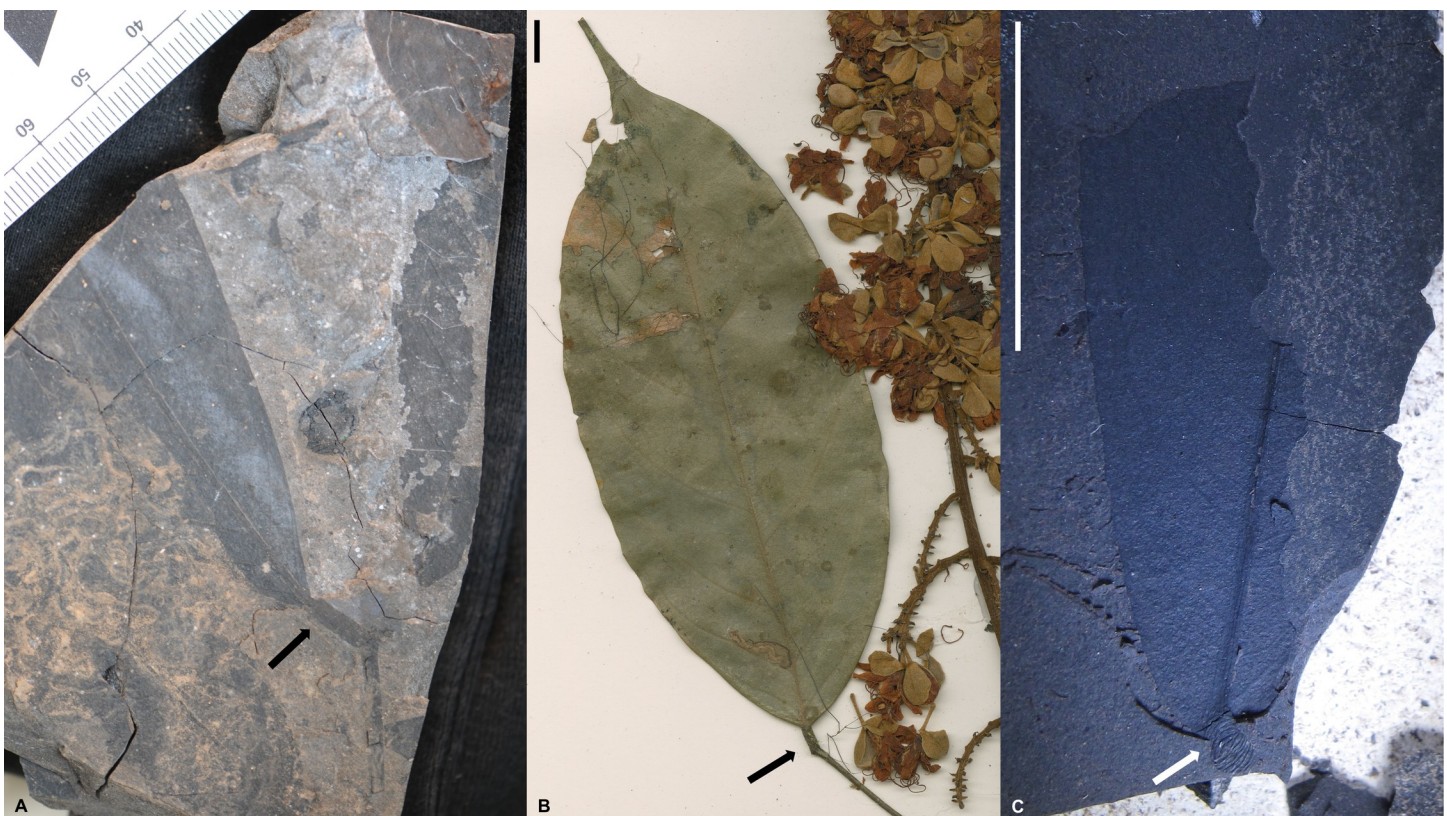

**Fig 1. Fossil and extant *Englerodendron* lamina.** A. *Englerodendron mulugetanum* sp. nov. paripinnate compound leaf (the twin leaflet is missing) with pulvinate petiolule. HOLOTYPE–MU17-43. B. Terminal leaflet of paripinnate leaf of *Englerodendron korupense* the twin leaflet is missing). ISOTYPE–MNHN-P-P00634905. Scale bar– 10 mm. Note twisted pulvinate petiolule. C. Leaflet of *Englerodendron mulugetanum* sp. nov. with detail of twisted pulvinate petiolule. MU27-18 #1. Scale bar– 10 mm.

length of the subjacent secondary, and have a frequency of < 1 per intercostal area (Fig 2). Tertiary venation is sinuous and generally opposite percurrent, rarely alternate percurrent (Fig 2). Fourth order veins are irregular reticulate, and fifth order veins freely ramify [22]. CUTICLE: Stomatal complexes are paracytic (incorrectly identified as pericytic in Bush et al., [22]) with one subsidiary cell larger than the other (Fig 3A). Stomata are prominent on the abaxial surface and are rare on the adaxial surface (Fig 3B). Guard cells range in size from 7.5 to 10.5 μm in length [23; Fig 2A]. The anticlinal cell walls on both the abaxial and adaxial leaf surfaces are highly sinuous [22], typically with greater than 9 'folds' and often numbering ≥11 (Fig 3A and 3B). The abaxial epidermal cells are rectangular in shape and are 21.3 to 32.7 μm in length and 9.8 to 14.1 μm in width [22; Fig 3A]. Papillae are absent from both abaxial and adaxial leaf epidermal periclinal cell surfaces, but knob-like thickenings appear to be present on the anticlinal walls (Fig 3A and 3B; [36]). Hairs (trichomes) are present on the abaxial surface, but sparse, based on presence of thickened trichome bases. Trichomes are commonly associated with the veins. Hair bases present in intercostal areas are surrounded by radial basal cells. Hair bases associated with veins are surrounded by vein epidermal cells oriented similarly to adjacent vein cells.

**Etymology:** The species name honors our friend and colleague, Dr. Mulugeta Feseha, of Addis Ababa University, who introduced us to the fossil locality. In addition, 'mulugeta' (ሙሉጌታ), which means 'he who rules' in Amharic, is particularly fitting since this species was an ecological and temporal dominant in the early Miocene Mush forest paleoenvironment.

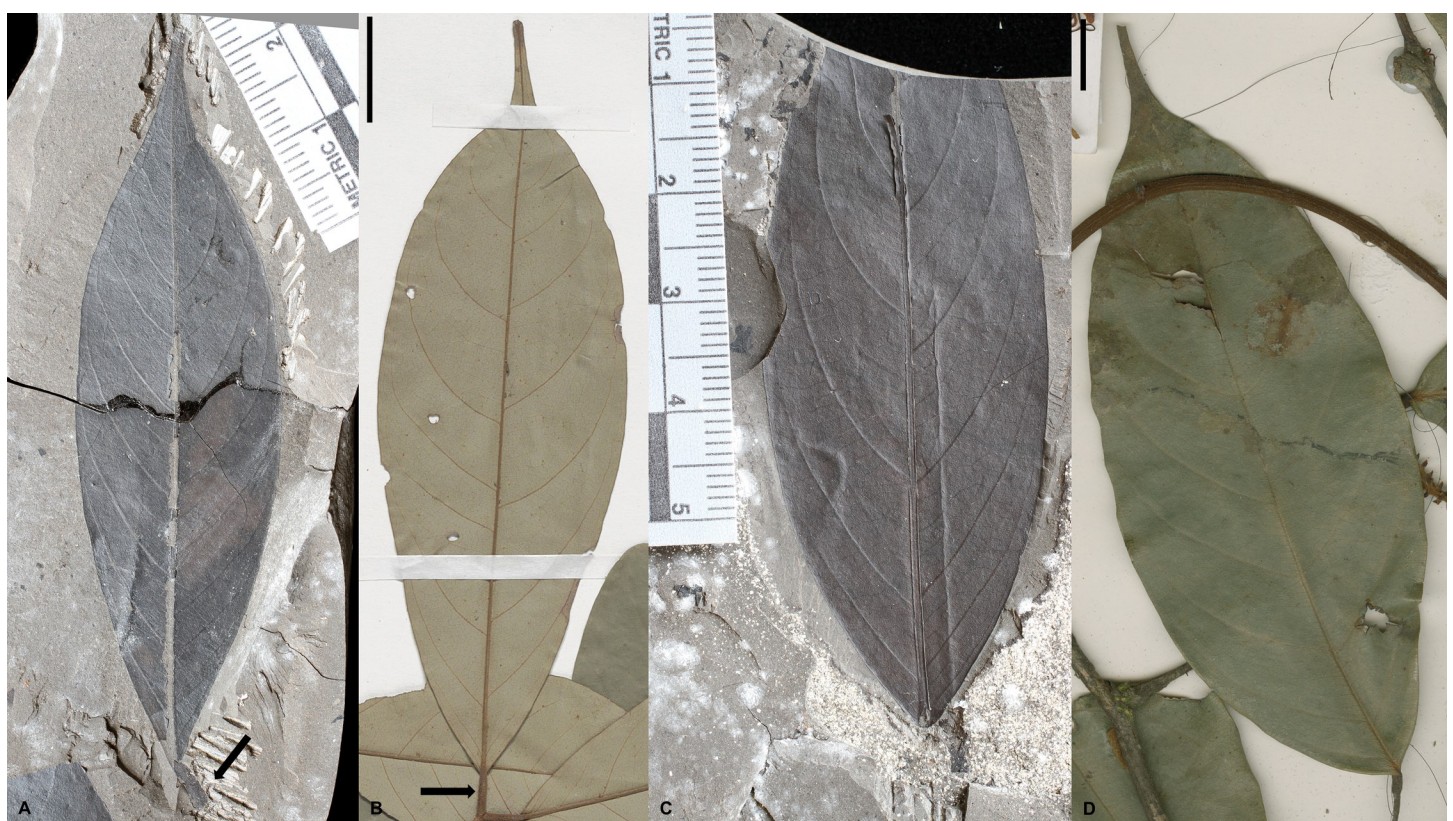

**Fig 2. Fossil and extant *Englerodendron* lamina continued.** A. *Englerodendron mulugetanum* sp. nov leaflet with a terete, pulvinate petiolule and secondary venation detail. MU29-40B #1. B. Leaflet of *Englerodendron vignei*. MNHN-P-P01037798. Scale bar– 10 mm. C. *Englerodendron mulugetanum* sp. nov. leaflet with detail of secondary, tertiary, and quaternary venation. MU29-41 #4. D. Leaflet of *Englerodendron korupense*. ISOTYPE–MNHN-P-P00634905. Scale bar– 10 mm.

**Comments:** Based on a suite of macro-morphological leaf characteristics, the fossil morphotype originally designated as "Legume 1" [22, 23] can be identified taxonomically as a member of the Berlinia Clade of the tribe Amherstieae (Leguminosae: Detarioideae), and specifically matches a number of genera in the Berlinia Clade: Subclade B, namely *Anthonotha*, *Berlinia*, *Englerodendron*, and *Isoberlinia* [37] (S5 and S6 Files). These characteristics include paripinnate compound leaves, oppositely inserted, symmetrical leaflets that are elliptic to slightly falcate in shape, the presence of a short twisted pulvinate petiolule, a symmetrical to slightly asymmetrical base, secondary venation being eucamptodromous becoming brochidodromous towards the apex, and an acuminate leaflet apex forming a drip tip [22, 38, 39] (Fig 1 and S5 and S6 Files) The fossil morphotype differs from *Oddoniodendron*, another member of Berlinia Clade Subclade B, in lacking a long petiolule and in not possessing alternately inserted leaflets [37, 40]. The African endemic moist forest genus *Gilbertiodendron* is similar to the fossil taxon and possesses many of the characteristics mentioned above, but differs from the Mush taxon in possessing leaflets with brochidodromous to festooned brochidodromous secondary venation and the presence of leaf marginal and/or submarginal glands [38, 39, 41, 42]. Other members in the Berlinia Clade, particularly genera in the 'bambjit' clade (*Brachystegia*, *Aphanocalyx*, *Michelsonia*, *Bikinia*, *Icuria*, *Tetraberlinia*) and *Microberlinia* possess asymmetrical leaflets [37, 38, 43] (S5 and S6 Files), have a major basal vein or 'fan' of veins, and often have retuse or emarginate leaflet apices, all of which are absent from the fossil taxon. These characteristics commonly occur in a number of other Amherstieae taxa as well, and so taxa

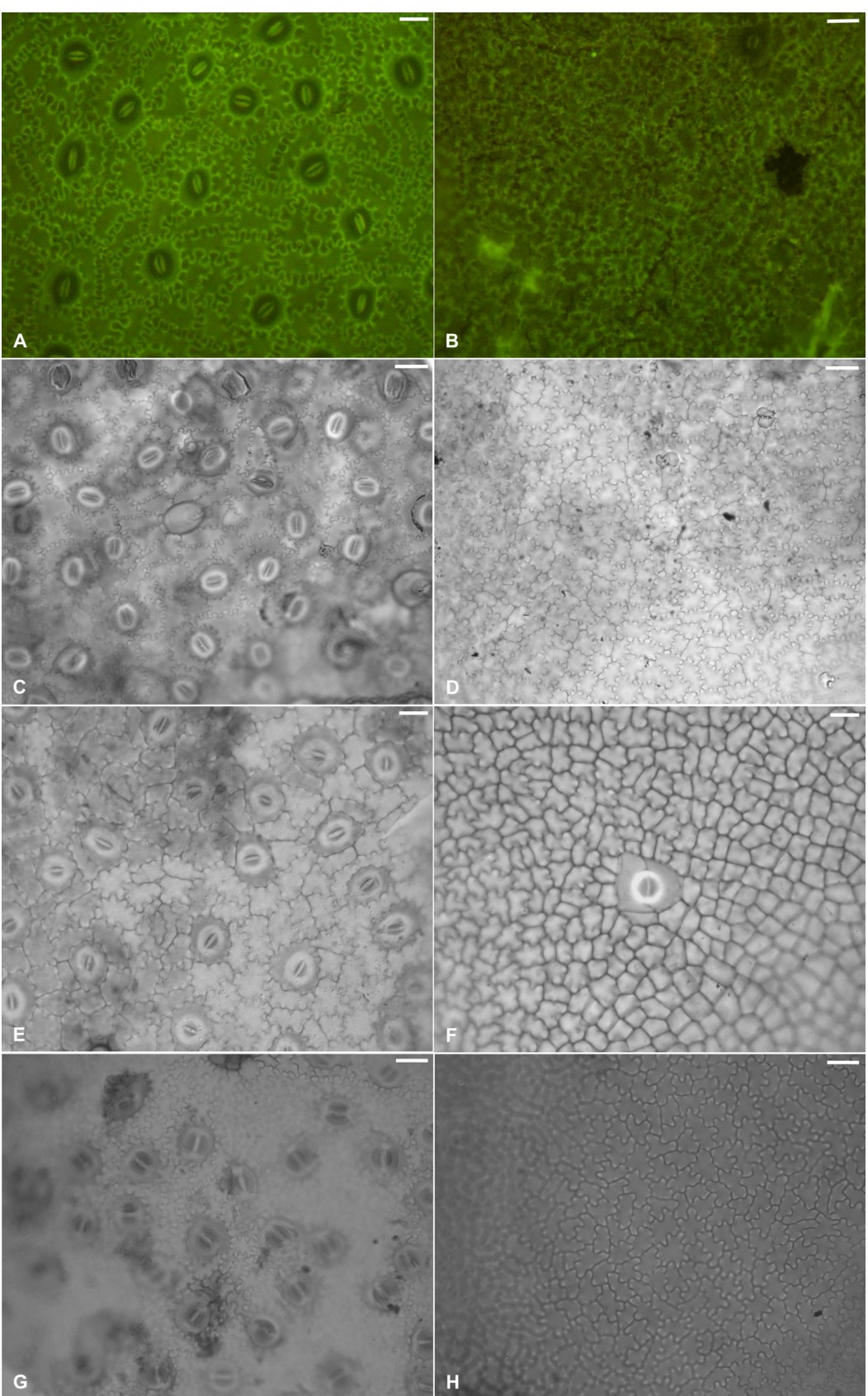

**Fig 3. Fossil and extant *Englerodendron* lamina cuticle.** Scale bars– 20 μm. A. *Englerodendron mulugetanum* sp. nov. abaxial leaflet surface. MU23-40(1C) #4. B. *Englerodendron mulugetanum* sp. nov. adaxial leaflet surface. MU41-15. C. *Englerodendron concyliophorum* abaxial leaflet surface. BM000081672. D. *Englerodendron concyliophorum* adaxial leaflet surface. BM000081672. E. *Englerodendron explicans* abaxial leaflet surface. MO2296034. F. *Englerodendron explicans* adaxial leaflet surface. MO2296034. G. *Englerodendron usambarensis* abaxial leaflet surface. K H/1194/96/56. H. *Englerodendron usambarensis* adaxial leaflet surface. K H/1194/96/56.

such as *Cynometra* clade A, *Cynometra* clade B, *Hymenostegia*, *Plagiosiphon*, *Scorodophloeus*, and *Talbotiella* can be eliminated as potential candidates for the fossil taxon [38, 44–46] (S5 and S6 Files). The presence of twisted pulvinate petiolules is a relatively rare character and can only be found in a handful of Amherstieae genera including *Crudia*, *Berlinia*, *Englerodendron*, *Gilbertiodendron*, *Librevillea*, *Oddoniodendron*, *Paramacrolobium*, and *Isoberlinia*. The fossil morphotype differs from *Crudia* leaflet symmetry, secondary venation (*Crudia* possesses brochidodromous/festooned brochidodromous venation), and leaves (alternately inserted leaflets in *Crudia*); *Librevillea*, in possessing alternately inserted, asymmetrical leaflets and imparipinnate leaves; *Paramacrolobium* in leaflet symmetry and lacking a major basal vein; and *Gilbertiodendron* and *Oddoniodendron* in the characteristics noted above [38] (S5 and S6 Files). It should be noted that *Gilbertiodendron*, *Berlinia*, *Englerodendron*, and *Isoberlinia* can possess both twisted and terete petiolules on the same plant or even compound leaf [38], S5 and S6 Files), unlike *Crudia*, where the character is constant. *Anthonotha*, *Berlinia*, *Englerodendron*, and *Isoberlinia* are the four genera having macro-morphological characteristics most similar to the fossil taxon, and among these, *Berlinia* and *Englerodendron* are the most closely comparable. *Isoberlinia* differs in typically possessing leaflets with noticeably asymmetrical leaflet bases and while leaflet apices may be acuminate (typically acute or rounded in savannah taxa), 'drip tips' are not present or not elongate.

The leaf epidermal cuticle characteristics allow the fossil to be placed in the genus *Englerodendron* as opposed to *Anthonotha*, *Berlinia*, and *Isoberlinia*. *Englerodendron*, and the more distantly related *Gilbertiodendron*, have highly sinuous anticlinal cell walls on both abaxial and adaxial leaf epidermal surfaces among species observed (Fig 2 and S7 File). The undulations on the anticlinal cell walls in species of these two genera are typically 'Ω' in shape (Fig 2 and S7 File). *Berlinia* possess sinuous anticlinal cell walls, but the undulations and fold amplitude are not as significant as in the fossil taxon, *Englerodendron*, or *Gilbertiodendron* and undulations are generally '∩' in shape [47] (S7 File). *Isoberlinia* anticlinal cell walls are also undulate, but not usually sinuous (although, *Isoberlinia scheffleri* can possess sinuous anticlinal cell walls; S7 File). *Anthonotha* differs from the fossil in being characteristically hairy on the abaxial leaf surface and in possessing papillae on abaxial periclinal cell surfaces (S7 File).

## Mush Morphotype (MT) pollen, affinity close to *Englerodendron conchyliophorum* and *E. korupense*

**Description:** Pollen, monad, prolate, P/E ratio 1.3–1.8, shape lobate-ellipsoid, elliptic in equatorial view, trilobate in polar view; equatorial diameter 17–42 μm in LM, 20–36 μm in SEM, polar axis 30–53 μm in LM, 35–61 μm in SEM; tricolporate; colpi long, extending between poles; pori inconspicuous, elongate elliptic(LM); exine 0.8–1.8 μm thick, nexine thinner than sexine (LM); tectate, colmellate (SEM); sculpture striate in LM, striate, perforate in SEM; aperture membrane granulate, nano- to microgemmate/areolate/rugulate, rugulate (SEM) (Fig 4 and Table 2).

**Comparison:** The Mush MT pollen shows a considerable size range when considering the equatorial diameter and length of the polar axis, also many of the grains are small. None of the pollen from the extant taxa shows such a wide size range nor are as small as the fossil pollen,

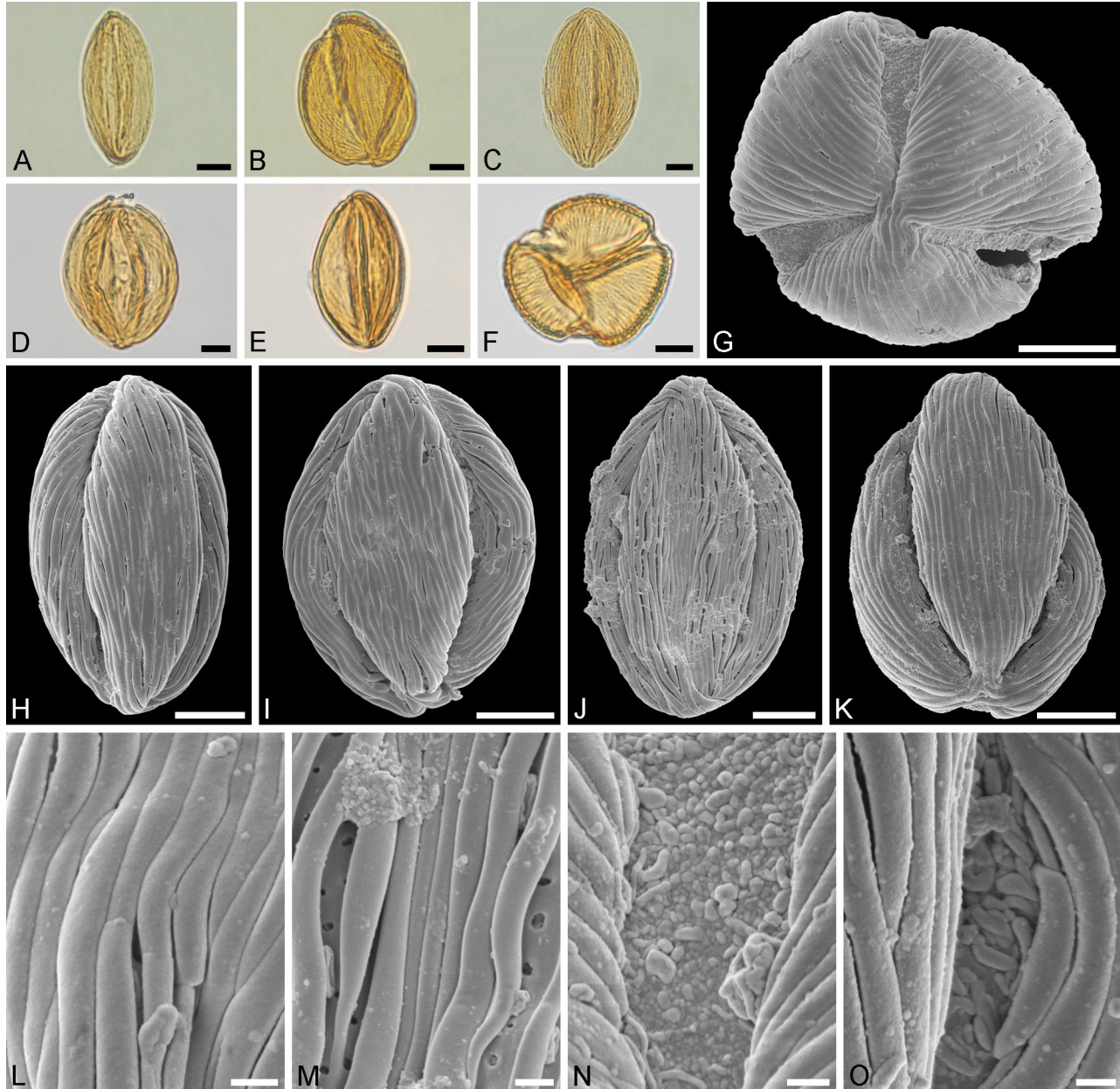

**Fig 4.** Light microscopy (A–F) and scanning electron microscopy (G–O) micrographs of the fossil Mush *Englerodendron* MT pollen, from Mush Valley, Ethiopia, Africa. A-E. Equatorial views, compressed pollen grains. F–G. Polar view, same compressed grain in LM and SEM. H–K. Equatorial views, compressed prolate pollen grains. L. Close-up of interapertural area, showing striate sculpture. M. Close-up of (J), showing interapertural area with striate and perforate sculpture. N. Close-up of (G), showing aperture with granulate and nano- to microgemmate/areolate sculpture membrane. O. Close-up of aperture, showing nanorugulate to regulate sculpture membrane. Scale bars– 10 μm (A–K), 1 μm (L–O).

but the Mush MT overlaps with most of them, and all size ranges measured for *Englerodendron explicans* pollen fall within the fossil MT. The Mush MT is clearly prolate, with a P/E ratio between 1.3–1.8, but pollen from most of the extant taxa are close to isodiametric, except for pollen of *E. conchyliophorum* and *E. korupense*, which are also prolate (Figs 3–5 and Table 2). The Mush MT is trilete in polar view and lobate-ellipsoid in shape like pollen from most of the extant taxa, except pollen of *Oddoniodendron micranthum*, which is hexagonal in polar view

**Table 2. Pollen morphology of fossil and extant Berlinia Clade pollen investigated for this study.**

| | Mush MT (fossil pollen) | Englerodendron conchyliophorum | Englerodendron explicans | Englerodendron korupense | Englerodendron leptorrhachis | Englerodendron mengei | Englerodendron usambarense | Berlinia orientalis | Isoberlinia scheffleri | Oddoniodendron micranthum |
|---|---|---|---|---|---|---|---|---|---|---|
| Equatorial diameter (LM) | 17–42 | 40–48 | 35–38 | 40–46 | 38–42 | 34–36 | 35–42 | 44–48 | 38–42 | 30–32 |
| Equatorial diameter (SEM) | 20–36 | 28–43 | 32–35 | 34–44 | 34–41 | 31–38 | 21–37 | 35–46 | 28–39 | 27–32 |
| Polar axis (LM) | 30–53 | 48–52 | 36–41 | 52–56 | 34–38 | 36–40 | 36–38 | 44–46 | 42–44 | 29–31 |
| Polar axis (SEM) | 35–61 | 47–62 | 41–47 | 51–60 | 33–39 | 34–45 | 36–46 | 39–51 | 42–49 | 26–30 |
| P/E ratio measured (LM) | 1.3–1.8 | 1–1.3 | 1–1.1 | 1.1–1.4 | 0.8–1 | 1–1.1 | 0.9–1.1 | 0.9–1 | 1–1.1 | 0.9–1 |
| P/E ratio category (LM) | Prolate | Isodiametric to prolate | Isodiametric to prolate | Prolate | Oblate to Isodiametric | Isodiametric to prolate | Oblate to prolate | Oblate to Isodiametric | Isodiametric to prolate | Oblate to isodiametric |
| Outline in equatorial view (LM) | Elliptic | Circular to elliptic | Circular to elliptic | Elliptic | Circular to elliptic | Circular to elliptic | Circular to elliptic | Circular | Circular to elliptic | Circular to elliptic |
| Outline in polar view (LM) | Trilobate | Trilobate | Trilobate | Trilobate | Circular (to trilobate) | Trilobate | Circular (to trilobate) | Circular to trilobate | Trilobate | Hexagonal |
| Pollen shape | Lobate-ellipsoid | Lobate-spheroid to ellipsoid | Lobate-spheroid to ellipsoid | Lobate-ellipsoid | Ellipsoid to spheroid | Lobate-spheroid to ellipsoid | (Lobate-)ellipsoid to spheroid | (Lobate-)ellipsoid to spheroid | Lobate-spheroid to ellipsoid | Hexagonal-dipyramid |
| Exine thickness (LM) | 0.8–1.8 | 1.6–2.6 | 1.8–2.4 | 1.6–2.4 | 1.2–1.6 | 1.8–2.4 | 1–2 | 2–3 | 1.6–2.4 | 1–1.6 |
| Sculpture LM | Striate | Striate | Striate | Striate | Striate | Striate | Striate | Striate | Striate | Rugulate |
| Sculpture SEM | Striate, perforate | Striate, perforate | Striate | Striate, perforate | Striate | Striate | Striate, perforate | Striate | Striate | Rugulate, verrucate, fossulate, perforate |
| Striae width (SEM) | 0.8–1.5 | 0.5–1.7 | 1.2–1.6 | 1.2–2 | 0.4–1.1 | 1.2–2 | 0.8–1.3 | 1.2–1.8 | 0.8–1.5 | N/A |
| Apertures | Tricolporate | Tricolporate | Tricolporate | Tricolporate | Tricolporate | Tricolporate | Tricolporate | Tricolporate | Tricolporate | Tricolporate |
| Visability pori (LM) | Inconspicuous | Inconspicuous | Inconspicuous | Conspicuous to inconspicuous | Conspicuous | Conspicuous | Conspicuous | Conspicuous | Conspicuous | Conspicuous |
| Outline pori (LM) | Lolongate elliptic | Lolongate elliptic | Lolongate elliptic | Lolongate elliptic | Lolongate elliptic | Lolongate elliptic | Circular to lolongate elliptic | Lolongate elliptic | Lolongate elliptic | Lalongate elliptic |
| Aperture membrane (SEM) | Granulate, nano- to microgemmate / areolate / rugulate, rugulate | Granulate, nano- to microrugulate | Nanorugulate to rugulate | Granulate, nanorugulate to rugulate | Granulate, nano- to microareolate / rugulate | Granulate, nano- to microrugulate | Granulate, nano- to microgemmate / areolate / rugulate | Nanorugulate to rugulate, granulate to areolate | Nano- to microrugulate, rugulate | Granulate, nano- to microgemmate / areolate |
| Overlapping features (LM & SEM) | N/A | 17 | 16 | 17 | 12 | 14 | 13 | 11 | 15 | 6 |

**Note:** All measurements include only those from this study and are given in µm. Features of extant pollen that overlap with the fossil Mush MT appear in bold font. N/A = not applicable.

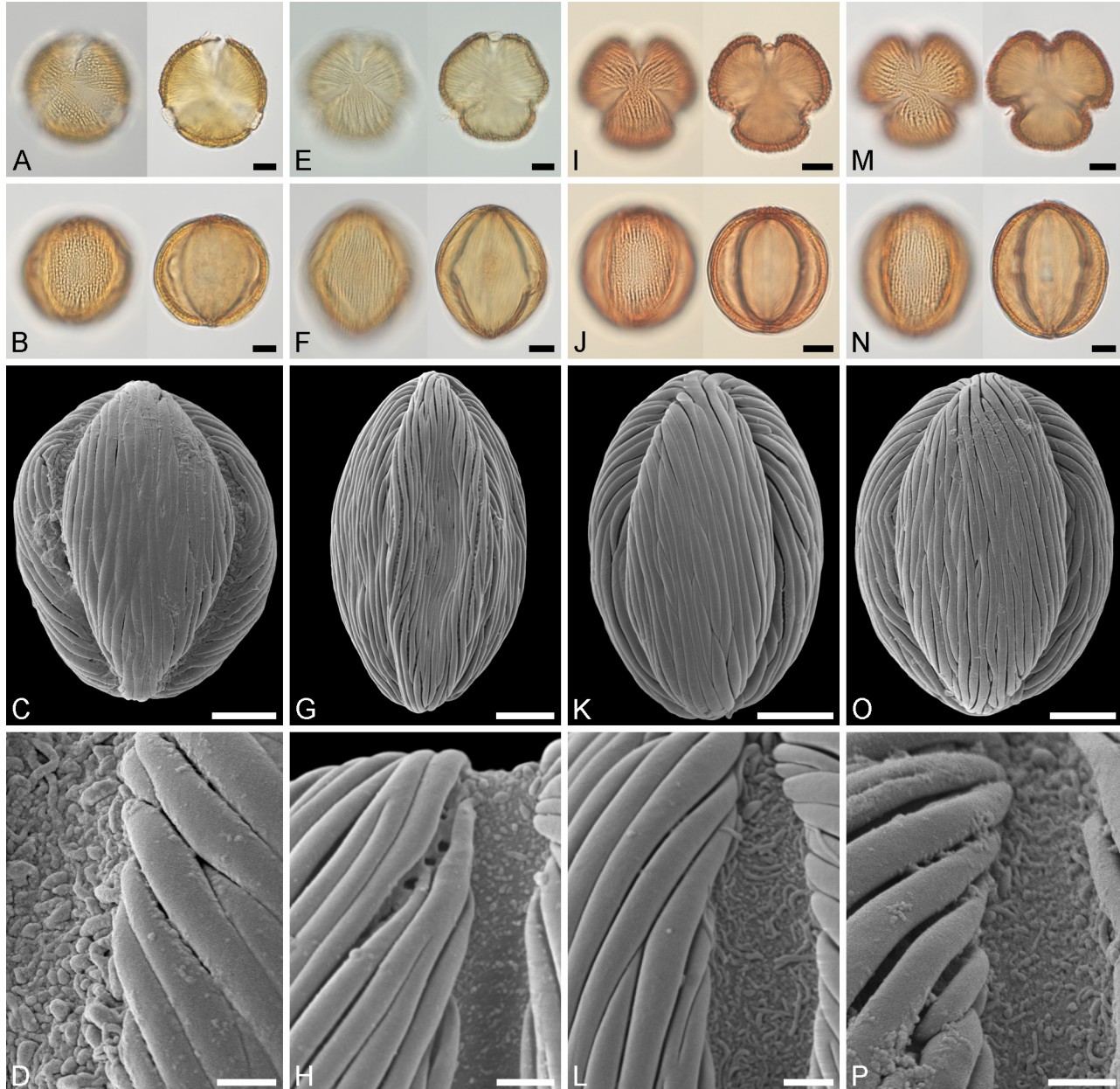

**Fig 5.** Light microscopy (A, B, E, F, I, J, M, N) and scanning electron microscopy (C, D, G, H, K, L, O, P) micrographs of extant Berlinia Clade pollen from Africa. A–D. *Berlinia orientalis* (from Tanzania, coll. FC Magogo & R Rose Innes, 409 [K000023099]). A. Polar views, high focus (left), optical cross-section (right). B. Equatorial views, high focus (left), optical cross-section (right). C. Equatorial view. D. Close-up showing aperture membrane and adjacent striate sculpture. E–H. *Englerodendron conchyliophorum* (from Cameroon, coll. s.n., s.n. [K000557101]). E. Polar views, high focus (left), optical cross-section (right). F. Equatorial views, high focus (left), optical cross-section (right). G. Equatorial view. H. Close-up showing aperture membrane and adjacent striate and perforate sculpture. I–L. *Englerodendron explicans* (from Guinea, coll. Haba, 1257 [K001381154]). I. Equatorial views, high focus (left), optical cross-section (right). J. Equatorial views, high focus (left), optical cross-section (right). K. Equatorial view. L. Close-up showing aperture membrane and adjacent striate sculpture. M–P. *Englerodendron korupense* (from Cameroon, coll. XM van der Burgt [K000264712]). M. Polar views, high focus (left), optical cross-section (right). N. Equatorial views, high focus (left), optical cross-section (right). O. Equatorial view. P. Close-up showing aperture membrane and adjacent striate sculpture. Scale bars– 10 μm (A–C, E–G, I–K, M–O), 2 μm (D, H, L, P).

and hexagonal-dipyramid in shape (Fig 4 and Table 2). The exine thickness of the Mush MT pollen displays a wide range, consistent with the size range of the morphotype (Table 2). The exine thickness overlaps with measurements from all the extant taxa, except *Berlinia orientalis*,

which has a considerably thicker exine (Table 2). The exine thickness of *Englerodendron leptor-rhachis* falls completely within the Mush MT, and the exine thickness of *E. usambarense* is also very close to that of the fossil MT. All pollen from extant *Englerodendron*, *Berlinia*, and *Isoberlina*, are striate when observed with LM, in the same way as the fossil MT, but pollen of *Oddoniodendron micranthum* is rugulate and clearly different when observed with either LM or SEM (Table 2). Only *Englerodendron conchyliophorum*, *E. korupense*, and *E. usambarense* show the perforate sculpture between the striae observed with SEM that is typical for the Mush MT. The width of the striae in the Mush MT overlaps with that of all investigated extant taxa except *Oddoniodendron*, which has a rugulate and verrucate sculpture when observed with SEM. The width of the striae in *Englerodendron conchyliophorum* and *E. usambarense* is close to that of the Mush MT, but the width of striae in *Isoberlinia scheffleri* is identical to that of the fossil. Both the Mush MT and pollen from all the investigated extant taxa is tricolporate, but the fossil pollen grains have inconspicuous pori that are hardly observed in either LM or SEM (Figs 4–6 and Table 2). This feature was also noticed for *Englerodendron conchyliophorum*, *E. explicans*, and *E. korupense* (Figs 4 and 5 and Table 2). The pori are lolongate elliptic in outline in the Mush MT and in pollen from most of the extant taxa, except *Oddoniodendron micranthum*, where the pollen is lalongate elliptic. The sculpture of the aperture membrane in the Mush MT is quite variable and overlaps with that observed in pollen from all the extant taxa. The combined features within each taxon shows that the Mush MT shares most morphological features with pollen of *Englerodendron conchyliophorum* (17 traits), *E. korupense* (17 traits), and *E. explicans* (16 traits), followed by *Isoberlinia scheffleri* (15 traits; Table 2).

**Comments:** The morphology (LM and SEM) of the fossil Mush MT pollen is similar to those of extant pollen produced by *Englerodendron*, *Isoberlinia*, and *Berlinia* (compared in Table 2). The Mush MT has a much broader size and P/E range and shows more variability in the general sculpture and in the sculpture of the aperture membrane than pollen from extant taxa. The reason for this is uncertain, but might be explained by one of the following: 1) the fossil pollen grains originate from the same biological taxon but reflect a much greater sampling source and originate from different anthers/flowers/individuals as compared to a single anther providing the extant comparison material. 2) the fossil pollen grains originate from more than a single biological taxon but their pollen overlap in morphological features and cannot be distinguished. 3) The fossil pollen grains originate from a single biological taxon and their morphological plasticity reflects an ancestral trait partly preserved in different extant descendants. Whatever the reason, it is clear that the Mush MT is most comparable to pollen from some extant *Englerodendron* and therefore supports a likely affiliation with the *Englerodendron mulugetanum* sp. nov. fossil leaves.

## Discussion

### Taxonomic, evolutionary, and biogeographic implications

*Englerodendron* is a moderately sized genus in the Amherstieae (Leguminosae: Detarioideae) consisting of 18 extant species belonging within a monophyletic subclade within the Berlinia Clade, referred to as Berlinia Clade Subclade B, which consists of *Anthonotha*, *Berlinia*, *Englerodendron*, *Isoberlinia*, *Librevillea*, and *Oddoniodendron* [48–52]. *Englerodendron* was formerly considered monospecific, consisting of a submontane forest species *E. usambarense*, endemic to the West Usambara Mountains in the Eastern Arc Mountains of Tanzania [53, 54]. The genus, which is sister to *Anthonotha*, now includes a number of species formerly included in the latter genus, as well as species of the now synonymized *Isomacrolobium* and *Pseudomacrolobium* [48, 50].

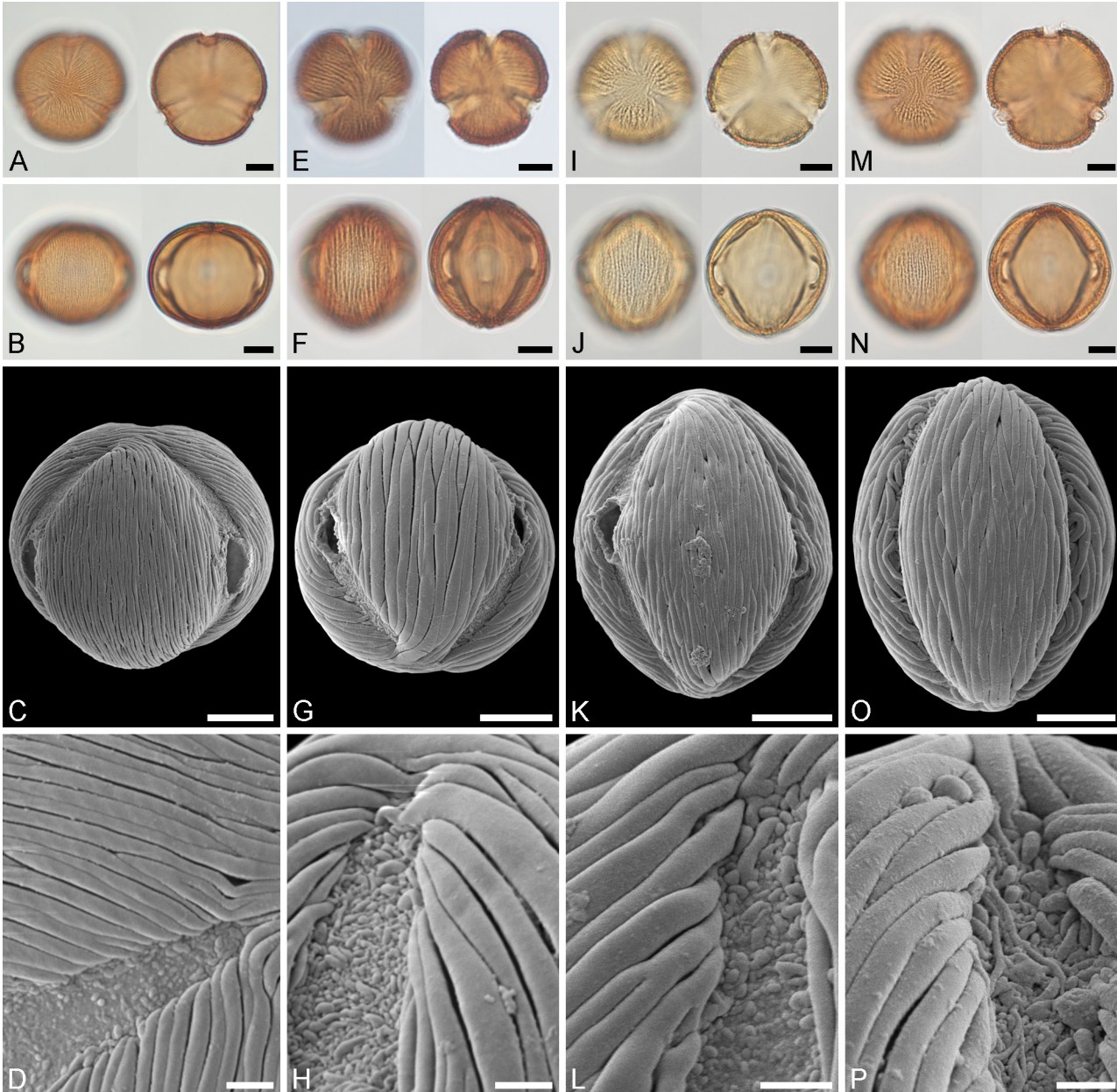

**Fig 6.** Light microscopy (A, B, E, F, I, J, M, N) and scanning electron microscopy (C, D, G, H, K, L, O, P) micrographs of extant Berlinia Clade pollen from Africa continued. A–D. *Englerodendron leptorrhachis* (from Cameroon, coll. Satabié, 980 [K001341298]). A. Polar views, high focus (left), optical cross-section (right). B. Equatorial views, high focus (left), optical cross-section (right). C. Equatorial view. D. Close-up showing aperture membrane and adjacent striate sculpture. E–H. *Englerodendron mengei* (from DR Congo, coll. J Louis, 3666 [K000557102]). E. Polar views, high focus (left), optical cross-section (right). F. Equatorial views, high focus (left), optical cross-section (right). G. Equatorial view. H. Close-up showing aperture membrane and adjacent striate sculpture. I–L. *Englerodendron usambarense* (from Tanzania, coll. Greenway, 1061 [K000555978]). I. Equatorial views, high focus (left), optical cross-section (right). J. Equatorial views, high focus (left), optical cross-section (right). K. Equatorial view. L. Close-up showing aperture membrane and adjacent striate sculpture. M–P. *Isoberlinia scheffleri* (from Tanzania, coll. Greenway, 3319 [K000557103]). M. Polar views, high focus (left), optical cross-section (right). N. Equatorial views, high focus (left), optical cross-section (right). O. Equatorial view. P. Close-up showing aperture membrane and adjacent striate sculpture. Scale bars– 10 μm (A–C, E–G, I–K, M–O), 2 μm (D, H, L, P).

*Englerodendron sensu lato* can now be characterized ecologically more broadly as a moist and wet forest genus (although *E. explicans* can also occur in savanna woodlands) that is mainly restricted to the Guineo-Congolian region along with the isolated Eastern Arc

*Englerodendron usambarense* [53, 54] (Fig 7). Species can occur in a variety of forest types including primary rain forest, semi-deciduous moist forests, riverine forest, gallery forest, submontane forest, and even lagoon-margin forest habitat [10, 48, 49, 52]. While extant species of *Englerodendron* do not develop into monodominant forest formations today, morphological and ecological characteristics hypothesized to assist in the development of such formations are prevalent within the genus and subclade. These characteristics include large seed size and short dispersal capabilities, shade-tolerant seedlings, ectomycorrhizal associations, conspecific gregariousness, and observations of coppicing within the genus are noted [2, 49, 55, 56].

The early Miocene *Englerodendron mulugetanum* sp. nov. is the only known fossil occurrence of the genus, although pollen attributed to closely related genera (*Anthonotha*, *Berlinia*, and *Isoberlinia*) is known from the late Paleocene and early Eocene of Nigeria and the late Oligocene—early Miocene of northern Kenya–indicating a long evolutionary history of the Berlinia Clade: Subclade B in Africa [37, 57–59]. Some fossil lamina impressions from the early Miocene Bugishu Series in Uganda also appear to be similar to *Berlinia*, but definitively assigning a generic identification to this material is not possible. These fossil records and the early Miocene (21.73 Ma) occurrence of *Englerodendron mulugetanum* sp. nov. indicate that late Pliocene or early Pleistocene evolutionary divergence time estimates for the genus and its sister taxon, *Anthonotha*, by Estrella et al. [37] need to be re-assessed.

In the Afrotropics, the majority of moist and wet monodominant forests consist of detarioid legume species as their principal constituent [4, 6, 60–64] (Table 1). Temporary monodominant forest formations dominated by non-legume taxa can develop in pioneer seral stage vegetation (e.g., *Musanga* spp.—Urticaceae; [3, 5, 65, 66]. Within the moist and wet detarioid legume African monodominant forest formations, all ecologically dominant species that fulfill this role belong within the Amherstieae, the largest and most diverse tribe in the Detarioideae

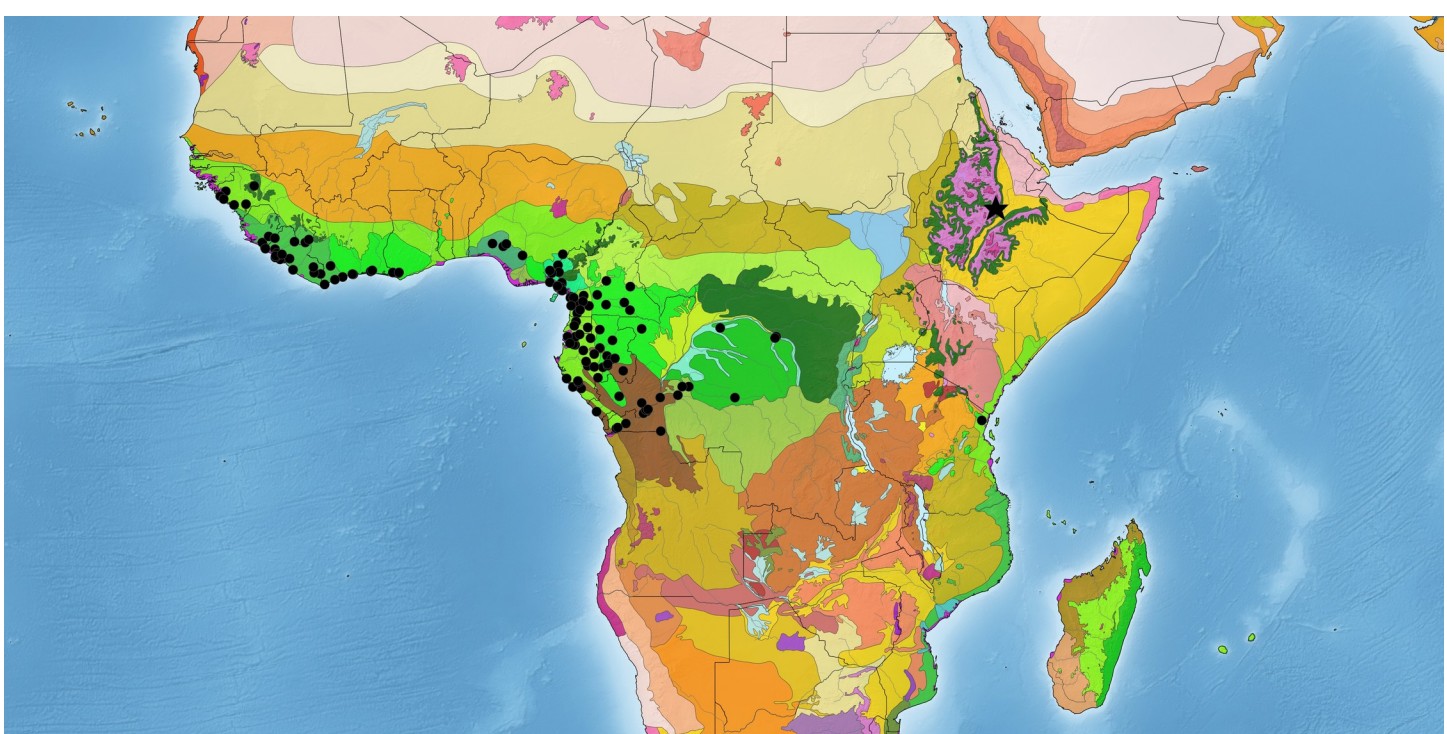

**Fig 7. Distribution map and terrestrial ecoregion occurrence of extant *Englerodendron* (circles) and location of the fossil species, *Englerodendron mulugetanum* (star).** This map was created with Simplemappr (https://www.simplemappr.net/).

composed of about 50 genera and about 570 species [44, 67]. Nevertheless, Subclade B of the Berlinia Clade of the tribe is absent among modern forest species dominants [37]. The majority of species that currently constitute monodominant forests in Africa are found in Subclade A of the Berlinia Clade, which is composed of the 'bambijt' clade + (*Gilbertiodendron + Didelotia*) [37]. The sole other species is found in the mid-level position in the phylogeny of the Amherstieae (*Cynometra alexandri*) [37]. Many genera within the Berlinia Clade of Amherstieae possess characteristics (as noted above) that could provide an ecological pathway to becoming an ecological dominant in moist and wet Afrotropical forests given an opportunity. Within the other speciose tribe within the Detarioideae, the Detarieae, multiple genera and species can be found in tropical African forest formations, but only one taxon, *Gilletiodendron glandulosum*, is known to dominate a forest community [68]. In this case, the community represents a relictual Sudanian-type of dry forest and falls outside the purview of this study [68]. Three additional Detarieae genera, *Colophospermum*, *Copaifera*, and *Guibourtia*, develop monotypic stands in African communities [10].

## Paleoecology and evidence for a monodominant forest

Extensive analyses of plant macro- and microfossils, and isotope and organic geochemistry from the Mush Valley lacustrine deposits provided a view of the surrounding community and the lake itself, which accumulated fine sediments and organic matter over some 50–60 ka [22, 23]. The following results of those studies together indicated the ancient terrestrial community was a forest:

1. The average leaf carbon isotope discrimination relative to atmospheric $CO_2$ ($\Delta^{13}C$) derived from organically preserved leaf compressions is 23.4 ‰, equal to that of modern tropical rainforests [22, 69].

2. The reconstructed mean annual precipitation among Levels A—F, ranges from $1523^{+199}_{-125}$ to $1638^{+215}_{-125}$ mm/yr, comparable to values supportive of forest vegetation [22].

3. Productive phytolith samples collected from 2–7 m above and below the macrofossil deposits are dominated by non-grass forest forms [23].

4. The $\delta^{13}C$ values of terrestrially sourced $C_{29}$ and $C_{31}$ *n*-alkanes (angiosperm dicots) from bulk organic samples among Levels A-F are −35 to −32‰, indicative of purely $C_3$ vegetation [23].

5. Terrestrial plant macrofossil identifications to this point include species limited to forest communities today: *Newtonia mushensis* [24], *Tacca umerii* [22–25, 27], and now *Englerodendron mulugetanum sp. nov.*

6. Terrestrial fossil pollen analyzed using combined LM and SEM, and even TEM, for single-grains also include taxa occurring in Africa's forests: *Aristogeitonia / Mischodon / Oldfieldia / Voatamalo* clade [70; fig. 19], *Sclerosperma* [71; fig. 3], and *Hagenia* [72; figs 6–8].

Although Currano et al. [23, 27] concluded the Mush forest was a mixed-moist, semi-evergreen closed-canopy forest similar to those found today in West, Central, and eastern Africa [e.g., 73], the long-term abundance and consistent dominance of *Englerodendron mulugetanum* sp. nov. leaflets at all lake levels sampled, and its placement in the Berlinia Clade, tribe Amherstieae, among which are species most commonly forming monodominant wet forests today, prompted our reconsideration of Mush forest structure. The consistent and overwhelming abundance of *E. mulugetanum* leaflets among the fossil assemblages is a compelling aspect of the collections, and led to the hypothesis of a monodominant forest

at Mush by Bush et al. [22]. This taxon comprises 53% of all classified leaves (1289 of 2427; leaflets are counted as leaves) and as much as 66% of specimens among the six stratigraphic levels ([23], and Table 3). Taphonomic processes, documented by studies of modern leaf fall, transport, litter accumulation, and preservation indicate that canopy leaves are more common than those from the understory in forest litter and local sedimentary deposits, and that leaves travel short distances before deposition, especially if thick and heavy; compound-leaved and deciduous trees can have an outsized impact on the composition of local (e.g., lakeside) deposits [74–77]. Thus, to understand results of these processes as they influence leaf accumulations from monodominant vs. mixed forests, litter studies of comparable modern tropical wet forests must be consulted. Analyses of modern leaf litter in African forests are rare, but Peh et al. [78] compared the litterfall, its rate of decomposition, and representation of the dominant taxon in litter samples in single-dominant vs. mixed forest plots in Cameroon (Table 3). Monodominant forests of *Gilbertiodendron dewevrei* (a Berlinia Clade legume) produced leaf litter whose dry weight was 52% - 92% (average of 73% ± 23%) *G. dewevrei* leaves [78]. Peh et al. [78] did not measure the proportion of leaf litter belonging to the most abundant species present in each mixed forested sampled, but the proportion of leaves belonging to the most abundant taxon in modern litter censuses from diverse Neotropical forests ranges from 12.8 to 31.4% (Table 3).

Measures of diversity are also lower for monodominant forests than mixed forests because of the presence of several species in low numbers (Table 3) [18, 63, 78]. In assessing diversity among the Mush leaf assemblages we chose to examine both rarefied species richness and Shannon diversity (Hill number, q = 2), which are commonly used in neoecological studies because values are more readily comparable among different communities [79]. A Hill number is the "effective number of species," which is defined as the number of equally abundant species needed to produce the given value of a diversity index [23, 79]. Rarefied richness gives

**Table 3. Comparison of diversity and dominance measures among fossil and modern localities [27, 63, 78].** Numbers shaded in blue highlight collections where the dominant taxon comprises ≥ 50%. Numbers shaded in green highlight leaf collections with Hill values ≥ 4.0.

| Locality | Age | Data Type | Number of Leaves | Species Richness at 300 Leaves | Hill # q = 2 | Percent Dominant Taxon |
|---|---|---|---|---|---|---|
| Guang | 27.1 Ma | Fossil leaves | 433 | 34.8 | **4.57** | 43.7 |
| Bull's Bellow | 27.1 Ma | Fossil leaves | 606 | 27.9 | **6.38** | 33.8 |
| Mush A | 21.7 Ma | Fossil leaves | 349 | 25 | **6.76** | 30.9 |
| Mush B | 21.7 Ma | Fossil leaves | 400 | 28.8 | 3.07 | **55.8** |
| Mush C | 21.7 Ma | Fossil leaves | 95 | NA | 3.97 | **49.5** |
| Mush D | 21.7 Ma | Fossil leaves | 532 | 30.9 | 3.50 | **50.9** |
| Mush E | 21.7 Ma | Fossil leaves | 510 | 27.0 | 3.12 | **55.7** |
| Mush F | 21.7 Ma | Fossil leaves | 541 | 20.4 | 2.16 | **65.8** |
| BCI | Modern | Leaf litter | 672 | 56.4 | **15.9** | 18.0 |
| Yasuni | Modern | Leaf litter | 951 | 50.7 | **7.9** | 31.4 |
| Rio Negro | Modern | Leaf litter | 932 | 70.1 | **22.1** | 12.8 |
| Peh mono1 | Modern | Leaf litter | | | | **52** |
| Peh mono2 | Modern | Leaf litter | | | | **73** |
| Peh mono3 | Modern | Leaf litter | | | | **92** |
| Kearsley monodominant | Modern | Tree plots, dbh data | | | 8.4 ± 1.2 | 65.3 |
| Kearsley mixed | Modern | Tree plots, dbh data | | | 21.2 ± 3.6 | 16.7 |

equal weight to rare and common taxa, whereas Shannon diversity emphasizes abundant taxa and can be thought of as the number of equally abundant common species. Table 3 provides diversity comparisons among Mush assemblages, modern leaf assemblages, and Kearsley et al.'s [63] study from monodominant and mixed forests in the Democratic Republic of Congo.

Fossil leaf collections from an Oligocene Ethiopian site (Chilga) to the north of Mush, and that have diversity measures demonstrating a richer and more even frequency distribution of taxa, provide a fossil point of comparison. The two Chilga localities listed in Table 3, Guang and Bull's Bellow, are contemporaneous, but located about 1.5 km apart. They share no taxa, and represent diverse, mixed forests at different stages of succession [80]. The most dominant taxon among leaves collected at each site comprises 44% and 34% of 433 and 606 leaves, respectively [27]. The Hill numbers are in line with, albeit lower than, leaf litter collections from a very diverse modern mixed forest at Yasuni, Ecuador [27].

As noted, *Englerodendron mulugetanum* sp. nov. is the most abundant taxon at each stratigraphic level collected at Mush. Level A, near the base of the section, contains only 31% *E. mulugetanum* sp. nov. leaflets and has a higher Shannon diversity than either Chilga site (Table 3 and Fig 8). Thus, we interpret the leaf assemblage from Level A as having come from a mixed rather than monodominant forest. Significant volcanic activity is associated with the formation of the paleolake, and it is possible that the forest at the time of Level A was on its way to monodominance. By the time of Level B, Shannon diversity drops significantly and *E. mulugetanum* sp. nov. leaflets comprise 55.8% of the assemblage [23], within the range that Peh et al. [78] observed in *G. dewevrei* monodominant forests (Fig 8). Shannon diversity remains low, and the proportion of *E. mulugetanum* sp. nov. leaflets high, for the remainder of the Mush stratigraphic levels (Fig 8). Perhaps the most compelling aspect of the Mush assemblages in support of monodominance is the consistent abundance of *E. mulugetanum* sp. nov. over the 50–60 kyrs sampled [23]. This fact argues against the presence of a single tree overarching or close to the shore of the ancient lake. In addition, although compound-leaved legume taxa are generally the most abundant overall, only *E. mulugetanum* sp. nov. dominates the assemblages at each stratigraphic level.

The presence of a long-lived, prehistoric monodominant forest in northwestern Ethiopia raises questions about the role of such forests in the evolutionary history and the

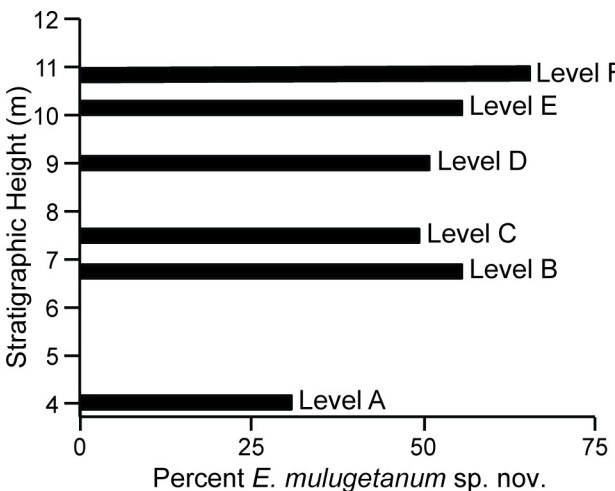

**Fig 8. Bar plot of *Englerodendron mulugetanum* sp. nov. abundance through the stratigraphic section.**

paleoecological processes of African forest vegetation during the early Miocene. Currently, no other Cenozoic tropical monodominant forests have been identified from before the Holocene. However, the likelihood that this assemblage represents the sole monodominant forest existing on the continent during the early Miocene is also relatively remote. As noted above, a number of monodominant forest types, differing in the dominant species constituent and type (persistence, seral stage, edaphically maintained, etc.), occur on the continent today and the only shared characteristic among them is that a single arborescent species makes up more than 50% of the canopy constituents of such forests [3, 81, 82]. Thus, monodominant forests do not harbor the magnitude of plant biodiversity and structural complexity as moist and wet mixed tropical forest formations, and consequently have bottom-up food web impacts for example, the diversity and intensity of frugivory and herbivory, as well as vertebrate biodiversity and biomass [83–86]. In most modern studies, observations of primate abundances and species-richness, elephant densities and activities, and rodent species-richness are reduced in *Gilbertiodendron dewevrei*-dominated monodominant forest formations in comparison to adjacent mixed and secondary forest formations [84, 85, 87, 88]. Typically, mammals prefer secondary and mixed forests due to the abundance of food resources in such forest types in comparison to primary or late secondary monodominant forests [83, 84, 89]. Correspondingly, the reduction of faunal elements in some forested habitats may actually increase the likelihood of monodominant forests developing or expanding [90]. There is some evidence that reduced elephant activity in western Uganda may assist in the development of *Cynometra alexandri* monodominant forests [91].

The presence of monodominant forests also creates additional vegetation heterogeneity within the tropical African sphere and how this affected forest faunal distributions and evolution is not known, but provides an additional aspect to consider–particularly in understanding dynamic changes in tropical forest distribution during the Miocene and how forest extent and composition may have affected particular mammal groups such as primates. The origin, temporal record, and spatial scale of Cenozoic monodominant forests in Africa are unknown, as Mush remains but a single datum point; however, this gap may be filled potentially by additional investigations of well-sampled, well-preserved plant macrofossil assemblages.

## Supporting information

**S1 File. Inclusivity in global research information and checklist.**
(DOCX)

**S2 File. Herbarium specimens examined.**
(XLSX)

**S3 File. Herbarium specimens sampled for micro-morphological (leaf cuticle) and palynological analysis.**
(XLSX)

**S4 File. Additional *Englerodendron mulugetanum* sp. nov. specimens.**
(PDF)

**S5 File. Table of comparing African Amherstieae (Leguminosae, Detarioideae) leaf and leaflet characteristics.**
(XLSX)

**S6 File. African Amherstieae (Leguminosae, Detarioideae) leaf and leaflet images and characteristics.**
(PDF)

**S7 File. Abaxial and adaxial leaf cuticle images of Berlinia Clade (Leguminosae, Detarioideae, Amherstieae) taxa.**
(PDF)

## Acknowledgments

We thank the Authority for Research and Conservation of Cultural Heritage for permission to conduct research in the Mush Valley and the director and staff of the National Museum of Ethiopia for facilitating our research. We are grateful to the people of Upper and Lower Mush for their hospitality, and to N. Tabor, M. Clemens, M. Feseha, L. Jacobs, J. Noret, D. Danehy, and T. Tesfamichael for contributions to field work. In particular we are grateful to M. Feseha for his role in bringing the Mush Valley locality to our attention. We are also grateful to D. Erwin, UC Museum of Paleontology, for providing images of fossils from the Bugishu Series in Uganda and T. Rehman, Botanical Research Institute of Texas, for providing herbarium material for us to examine. We are also thankful to the two anonymous reviewers whose comments improved this work.

## Author Contributions

**Conceptualization:** Aaron D. Pan, Bonnie F. Jacobs, Friðgeir Grímsson, Ellen D. Currano.

**Data curation:** Aaron D. Pan, Bonnie F. Jacobs, Rosemary T. Bush, Friðgeir Grímsson, Patrick S. Herendeen, Ellen D. Currano.

**Formal analysis:** Aaron D. Pan, Bonnie F. Jacobs, Rosemary T. Bush, Friðgeir Grímsson, Ellen D. Currano.

**Funding acquisition:** Bonnie F. Jacobs, Friðgeir Grímsson, Ellen D. Currano.

**Investigation:** Aaron D. Pan, Bonnie F. Jacobs, Friðgeir Grímsson, Patrick S. Herendeen, Ellen D. Currano.

**Methodology:** Aaron D. Pan, Bonnie F. Jacobs, Friðgeir Grímsson, Ellen D. Currano.

**Project administration:** Aaron D. Pan, Bonnie F. Jacobs, Ellen D. Currano.

**Resources:** Aaron D. Pan, Bonnie F. Jacobs, Friðgeir Grímsson, Ellen D. Currano.

**Software:** Aaron D. Pan, Bonnie F. Jacobs, Ellen D. Currano.

**Supervision:** Aaron D. Pan, Bonnie F. Jacobs, Ellen D. Currano.

**Validation:** Aaron D. Pan, Bonnie F. Jacobs, Rosemary T. Bush, Manuel de la Estrella, Friðgeir Grímsson, Patrick S. Herendeen, Xander M. van der Burgt, Ellen D. Currano.

**Visualization:** Aaron D. Pan, Bonnie F. Jacobs, Friðgeir Grímsson, Patrick S. Herendeen, Ellen D. Currano.

**Writing – original draft:** Aaron D. Pan, Bonnie F. Jacobs, Rosemary T. Bush, Manuel de la Estrella, Friðgeir Grímsson, Patrick S. Herendeen, Xander M. van der Burgt, Ellen D. Currano.

**Writing – review & editing:** Aaron D. Pan, Bonnie F. Jacobs, Rosemary T. Bush, Manuel de la Estrella, Friðgeir Grímsson, Patrick S. Herendeen, Xander M. van der Burgt, Ellen D. Currano.

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
