## [Decision Letter · Decision Letter 0]

28 Oct 2022

PONE-D-22-26695FIRST EVIDENCE OF A MONODOMINANT (ENGLERODENDRON, AMHERSTIEAE, DETARIOIDEAE, LEGUMINOSAE) TROPICAL MOIST FOREST FROM THE EARLY MIOCENE (21.73 MA) OF ETHIOPIAPLOS ONE

Dear Dr. Pan,

Thank you for submitting your manuscript to PLOS ONE. After careful consideration, we feel that it has merit but does not fully meet PLOS ONE’s publication criteria as it currently stands. Therefore, we invite you to submit a revised version of the manuscript that addresses the points raised during the review process.

We look forward to receiving your revised manuscript.

Kind regards,

Gongle Shi, Ph.D.

Academic Editor

PLOS ONE

Journal Requirements:

4. We note that Figure 6 in your submission contain [map/satellite] images which may be copyrighted. All PLOS content is published under the Creative Commons Attribution License (CC BY 4.0), which means that the manuscript, images, and Supporting Information files will be freely available online, and any third party is permitted to access, download, copy, distribute, and use these materials in any way, even commercially, with proper attribution. For these reasons, we cannot publish previously copyrighted maps or satellite images created using proprietary data, such as Google software (Google Maps, Street View, and Earth). For more information, see our copyright guidelines: http://journals.plos.org/plosone/s/licenses-and-copyright.

a. You may seek permission from the original copyright holder of Figure 6 to publish the content specifically under the CC BY 4.0 license.  

5. Please take this opportunity to be sure you have met all of our guidelines for new species. For submissions describing new species that do not have formal registries, please include a sub-section called “Nomenclature” in the Methods section using the following wording:

The electronic version of this article in Portable Document Format (PDF) in a work with an ISSN or ISBN will represent a published work according to the International Code of Nomenclature for algae, fungi, and plants, and hence the new names contained in the electronic publication of a PLOS ONE article are effectively published under that Code from the electronic edition alone, so there is no longer any need to provide printed copies.

The online version of this work is archived and available from the following digital repositories: PubMed Central, LOCKSS [author to insert names of any additional repositories where the work will be deposited].

Reviewers' comments:

Reviewer's Responses to Questions

**Comments to the Author**

1. Is the manuscript technically sound, and do the data support the conclusions?

Reviewer #1: Yes

Reviewer #2: Yes

2. Has the statistical analysis been performed appropriately and rigorously? 

Reviewer #1: Yes

Reviewer #2: Yes

3. Have the authors made all data underlying the findings in their manuscript fully available?

Reviewer #1: Yes

Reviewer #2: Yes

4. Is the manuscript presented in an intelligible fashion and written in standard English?

Reviewer #1: Yes

Reviewer #2: Yes

5. Review Comments to the Author

Reviewer #1: This manuscript describes an exciting association of fossil leaflets of Englerodendron mulugeta sp. nov., from the Early Miocene of Ethiopia. The work is important for providing evidence of a monodominant tropical moist forest. I applaud the authors for compiling large and valuable images/datasets of cuticular and pollen information. I particularly like the quality of the SEM images of the pollen.

I have carefully reviewed the manuscript (see marked pdf), and here I provide some points:

The tiff files of figures 1 & 2 I was able to download are not of the best resolution. I had difficulty seeing some characters mentioned in the diagnosis and detailed description. I also would like to see additional closeups of some of the leaflets (e.g., twisted pulvinus, secondary and higher order venation.) Since this is the flora's most dominant taxon, there should be plenty of images. It looks like this "Legume 1 = Englerodendron mulugeta sp. nov" has been informally described/figured in other articles. However, the reader of this paper should see all the relevant images given that a new species is being erected.

The images of the fossil cuticles of the new species appear blurry in the Tiff file. Please consider new photos of the fossil cuticles at 10um to facilitate the comparisons with living taxa.

I have a few concerns regarding the monodominance of the forest, and I like that the authors stated, "it is possible that the forest at the time of Level A was on its way to monodominance". Although the percentages of the fossil leaflets are relatively large, they don't appear to be entirely the dominant taxon when compared with the few data available from tropical Africa (Table 3, Peh mono1-3). Considering that the leaflets were counted as leaves needs additional treatment in the statistical analyses.

It would be nice if the coauthors added a new figure plotting "Percent Dominant Taxon" against the stratigraphic levels.

Reviewer #2: This manuscript describes Englerodendron mulugeta sp. nov., a Detarioid legume from the early Miocene of Ethiopia based on abundant leaflets recovered from the Mush Valley site. The authors compare the fossil leaves with extant species of Detarioideae to support their taxonomic assessment and provide evidence of fossilized pollen that may correspond to the same or closely related species. The high abundance of leaves and pollen throughout the 50–60kyr sequence of the Mush Valley site are interpreted as evidence of a monodominant forest community dominated by a tree species belonging to a linage in which ecological dominants are common.

The manuscript is clear, the methodology is sound and the results are well-supported. I recommend this paper for publication. There are only three minor suggestions that I consider the authors should address:

1. Please revise the international code of botanical nomenclature for correct epithet assignation. Is ‘mulugeta’ treated as a latinized noun in apposition? The reference that the authors provide for this epithet is “he who rules”, which sounds more like an adjective instead. If this is the case (adjective), the adequate spelling should be “mulugetum”. If naming in honor of Dr. Mulugeta Fesesha it should be “mulugetanum”.

2. Figure 1 – The quality of images is quite poor and therefore the species description and morphological comparisons are hard to follow. The little detail that was visible is consistent with the species description, but I suggest that the authors include better images and/or consider including an illustration of the type specimen. Particularly the fine venation is impossible to figure out.

3. Even though the authors include a Table with detailed morphological comparison between the fossils and living species of Detarioideae, I encourage the authors to include a supplementary figure with leaves of the species they compared the fossils with, as a means to provide stronger evidence for their taxonomic assessment.

6. PLOS authors have the option to publish the peer review history of their article (what does this mean?). If published, this will include your full peer review and any attached files.

Reviewer #1: No

Reviewer #2: No

---

## [Author Response · Author response to Decision Letter 0]

6 Nov 2022

Dr. Gongle Shi, Academic Editor, and Reviewer 1 and Reviewer 2:

Thank you very much for your review of our manuscript [PONE-D-22-26695]: First evidence of a monodominant (Englerodendron, Amherstieae, Detarioideae, Leguminosae) tropical moist forest from the early Miocene (21.73 Ma) of Ethiopia, on behalf of PLOS ONE. We appreciate the opportunity to submit the minor revision of our paper for your consideration. Below we respond to the editor’s and reviewers’ comments and note corresponding changes we have made in our paper.

We have reviewed our revised manuscript to make sure that it meets PLOS ONE’s style requirements. We have included a completed copy of PLOS’ questionnaire on inclusivity in global research with the revised manuscript and have included it in Supporting Information as S1. We have included our full ethics statement in the Methods section. 

PLOS ONE and the academic editor had some concern about Figure 6, now Figure 7 in the revised manuscript, and that it might copyrighted. This figure was created solely using Simplemappr, which is in the public domain (similar to USGS National Map Viewer, The Gateway to Astronaut Photography of Earth, etc.) and which we have seen used in other article published by PLOS ONE. The following language is included on the website: “All versions of SimpleMappr map data found on this website are in the Public Domain. You may use the maps in any manner, including modifying the content and design, electronic dissemination, and offset printing. The primary author, David P. Shorthouse has waived all copyright, related or neighboring rights, and financial claim to the maps and invites you to use them for personal, educational, and commercial purposes. No permission is needed to use SimpleMappr. Crediting the author is unnecessary.” (https://www.simplemappr.net/#tabs=6). In the figure legend we have included the following language, “This map was created with Simplemappr (https://www.simplemappr.net/).” 

We have included the language required for new species in a “Nomenclature” sub-section in the Methods section as required by PLOS ONE. In addition, we have changed all ‘Fabaceae’ to ‘Leguminosae’ for consistency. In addition, small grammatical or punctuation errors that we have found in the course of this revision have been corrected.

We have reviewed the reference section and have made no changes. We have updated some information in the Supporting Information to provide additional images as requested by Reviewer 2. 

Below you will find our responses to the comments and concerns of Reviewer 1 and Reviewer 2:

Reviewer 1 (R1) concern:

“The tiff files of figures 1 &2 I was able to download are not of the best resolution. I had difficulty seeing some characters mentioned in the diagnosis and detailed description. I would also like to see additional closeups of some of the leaflets (e.g., twisted pulvinus, secondary and higher order venation.) Since this is the flora’s most dominant taxon, there should be plenty of images. It looks like this “Legume 1 – Englerodendron mulugeta sp. nov.” has been informally described/figured in other articles. However, the reader of this paper should see all the relevant images given that a new species is being erected.”

Authors’ Response:

We have redone Fig 1 and have made an additional figure (which is now Fig 2) so that the characters in the diagnosis and description can be more easily seen. This includes high resolution and larger images of the fossil species, as well as extant species. We have specifically added an image of MU29-41 #4 (Figure 2C), which has very good details of the canaliculate midvein, and the secondary, tertiary, and quaternary venation. 5th level venation of the taxon can be observed in Figure 3F of Bush et al. (2017). In addition, in the Supporting information we have provided images of 4 more specimens that provide additional viewing of the characteristics noted in the manuscript.

Reviewer 1 (R1) concern:

“The images of the fossil cuticles of the new species appear blurry in the Tiff file. Please consider new photos of the fossil cuticles at 10 μm to facilitate the comparisons with living taxa.”

Authors’ Response:

We have selected new images of the fossil cuticle and have made sure that the resolution is at 300 dpi.

Reviewer 1 (R1) concern:

“I have a few concerns regarding the monodominance of the forest, and I like that the authors stated, “it is possible that the forest at the time of Level A was on its ways to monodominance”. Although the percentages of the fossil leaflets are relatively large, they don’t appear to be entirely the dominant taxon when compared with the few data available from tropical Africa (Table 3, Peh mono 1-3). Considering that the leaflets were counted as leaves needs additional treatment in the statistical analyses.”

Authors’ Response:

The dominance of the fossil Englerodendron in the Mush plant assemblages is particularly notable, not only in terms of the abundance of the morphotype/taxon in Levels B - F, and prevalence in Level A, but also in representing the most abundant taxon at around 5 – 6.5 times (excluding Level A, which is 3 times) the level of the next most abundant species over a duration of tens of thousands of years. The taxon does not appear to be a short-term proliferating species that has been sampled in a single large stratum in abundance, but in multiple levels, likely showing an estimated 40,000 – 50,000 years of dominance. In regards to comparisons with Peh et al.’s (2012) study, we believe that it is valid. The dominant species in his work is Gilbertiodendron dewevrei, which is also a paripinnate-compound leaved legume species in the Berlinia Clade. Gilbertiodendron dewevrei, is typically 3-jugate, so each leaf usually consists of 6 leaflets. In the leaf litter fall studies whole and partial leaves (including leaflets) are used in litterfall weighing. In addition, the new fossil Englerodendron species is not the only compound-leaved legume in the flora, with the next two taxa (‘Short-drip tip’ and ‘Interrupt’) each making up 4 – 10% of the material in each level, but never to the extent of the fossil Englerodendron species. Conversely, other taxa present in the fossil flora, Newtonia mushensis (Pan et al. 2012) and Zanthoxylum sp., are only represented as fruits and seeds. Both genera, Newtonia (Leguminosae: Mimoseae) and Zanthoxylum (Rutaceae), are noted for possessing diagnostic leaves (with diagnostic leaflets) that are pinnately-compound. No potential candidates representing either of these genera have been found amongst the fossil leaf(let) material from Mush. We believe the statistical analyses in which we have used Hill numbers for Mush and Chilga (Guang and Bull’s Bellow) provide evidence for the presence of monodominance of Englerodendron at Mush. We also note that the dominant taxa at Chilga are Cynometra chaka (Guang) and an Albizia-like legume leaf type (Bull's Bellow), both of which have compound leaves.

Reviewer 1 (R1) concern:

“It would be nice if the coauthors added a new figure plotting “Percent Dominant Taxon” against the stratigraphic levels.”

Authors’ Response: 

We have added a bar graph figure, Fig 8, which shows percent of abundance of the fossil Englerodendron leaflets plotted against stratigraphic height. In addition, similar bar plots for abundances of the taxa can be found in Currano et al. (2020) as Figure 6.

In addition, we have made almost all of the recommended changes that Reviewer 1 (R1) noted in the manuscript. We have kept the description of some of the petioles as ‘terete’ instead of ‘interpreted or likely terete’ as suggested by R1, because some of the fossils, particularly as seen in Fig 2A and S5, show straight pulvinate petioles indicating the presence of this characteristic, along with some specimens possessing twisted pulvinate petiolules in the fossil Englerodendron species.

Reviewer 2 (R2) concern:

“1. Please revise the international code of botanical nomenclature for correct epithet assignation. Is ‘mulugeta’ treated as a latinized noun in apposition? The reference that the authors provide for this epithet is “he who rules”, which sounds more like an adjective instead. If this is the case (adjective), the adequate spelling should be “mulugetum”. If naming in honor of Dr. Mulugeta Fesesha it should be “mulugetanum”.”

Authors’ Response: 

We fully agree and thank R2 for informing us on this issue. We have changed the proposed species name to Englerodendron mulugetanum and have changed the etymology section accordingly.

Reviewer 2 (R2) concern:

“2. Figure 1 – The quality of images is quite poor and therefore the species description and morphological comparisons are hard to follow. The little detail that was visible is consistent with the species description, but I suggest that the authors include better images and/or consider including an illustration of the type specimen. Particularly the fine venation is impossible to figure out.”

Authors’ Response: 

This is similar to the concerns of R1 and we have changed the images accordingly to make sure that we have provided images to assist the reader in recognizing and viewing the characteristics of the new species. We have redone Fig 1 and added another figure (now Fig 2) so that the characters in the diagnosis and description can be determined more easily, including high resolution images of the fossil and extant species. Specimen MU29-41 #4 (Figure 2C) has been added and provides very good details of the canaliculate midvein, and the secondary, tertiary, and quaternary venation. 5th level venation of the taxon can be observed in Figure 3F of Bush et al. (2017). We have also provided images of 4 additional specimens in Supporting information.

Reviewer 2 (R2) concern:

“3. Even though the authors include a Table with detailed morphological comparison between the fossils and living species of Detarioideae, I encourage the authors to include a supplementary figure with leaves of the species they compared the fossils with, as a means to provide stronger evidence for their taxonomic assessment.”

Authors’ Response: 

We have created an additional document in Supporting Information that includes a number of examples of other Amherstieae leaf(let) images and characteristics to assist the reader. This was a very good idea. Thank you!

Again, we want to thank the academic editor and the 2 reviewers for very helpful suggestions to assist in making this a better paper. If you have any additional questions or need more information, please let us know. Thank you again for consideration of our revised manuscript.

Kind regards,

Aaron D. Pan, Ph.D.

Executive Director

Museum of Texas Tech University

Texas Tech University

---

## [Editor Report · Decision Letter 1]

8 Dec 2022

FIRST EVIDENCE OF A MONODOMINANT (ENGLERODENDRON, AMHERSTIEAE, DETARIOIDEAE, LEGUMINOSAE) TROPICAL MOIST FOREST FROM THE EARLY MIOCENE (21.73 MA) OF ETHIOPIA

PONE-D-22-26695R1

Dear Dr. Pan,

We’re pleased to inform you that your manuscript has been judged scientifically suitable for publication and will be formally accepted for publication once it meets all outstanding technical requirements.

Kind regards,

Gongle Shi, Ph.D.

Academic Editor

PLOS ONE
---

## [Editor Report · Acceptance letter]

14 Dec 2022

PONE-D-22-26695R1 

First evidence of a monodominant (*Englerodendron*, Amherstieae, Detarioideae, Leguminosae) tropical moist forest from the early Miocene (21.73 Ma) of Ethiopia 

Dear Dr. Pan:

I'm pleased to inform you that your manuscript has been deemed suitable for publication in PLOS ONE. Congratulations! Your manuscript is now with our production department. 

Kind regards, 

on behalf of

Dr. Gongle Shi 

Academic Editor

PLOS ONE